# Partial Label Learning with Dissimilarity Propagation guided Candidate Label Shrinkage

**Yuheng Jia[1,3]***, **Fuchao Yang[2]**, **Yongqiang Dong[1]***

[1] School of Computer Science and Engineering, Southeast University, Nanjing 210096, China
[2] College of Software Engineering, Southeast University Nanjing 210096, China
[3] Key Laboratory of New Generation Artificial Intelligence Technology and Its
Interdisciplinary Applications (Southeast University), Ministry of Education, China
yhjia@seu.edu.cn, yangfc@seu.edu.cn, dongyq@seu.edu.cn

## Abstract

In partial label learning (PLL), each sample is associated with a group of candidate labels, among which only one label is correct. The key of PLL is to disambiguate the candidate label set to find the ground-truth label. To this end, we first construct a constrained regression model to capture the confidence of the candidate labels, and multiply the label confidence matrix by its transpose to build a second-order similarity matrix, whose elements indicate the pairwise similarity relationships of samples globally. Then we develop a semantic dissimilarity matrix by considering the complement of the intersection of the candidate label set, and further propagate the initial dissimilarity relationships to the whole data set by leveraging the local geometric structure of samples. The similarity and dissimilarity matrices form an *adversarial* relationship, which is further utilized to shrink the solution space of the label confidence matrix and promote the dissimilarity matrix. We finally extend the proposed model to a kernel version to exploit the non-linear structure of samples and solve the proposed model by the inexact augmented Lagrange multiplier method. By exploiting the adversarial prior, the proposed method can significantly outperform state-of-the-art PLL algorithms when evaluated on 10 artificial and 7 real-world partial label data sets. We also prove the effectiveness of our method with some theoretical guarantees. The code is publicly available at `https://github.com/Yangfc-ML/DPCLS`.

## 1 Introduction

Partial label learning (PLL) [14, 15, 17] is an emerging weakly supervised learning framework. In PLL, each sample is associated with a set of candidate labels, among which only one is the ground-truth label. Different from conventional supervised learning, PLL avoids precisely annotating label on each sample, which greatly reduces the labeling cost. Accordingly, PLL has been applied to many real-world scenarios such as automatic image annotation [1, 20], web mining [9], and ecoinformatics [8].

Formally speaking, let $\mathcal{X} = \mathbb{R}^d$ be the $d$-dimensional feature space and $\mathcal{Y} = \{1, 2, ..., q\}$ be the label space with $q$ labels. Given the partial label training set $\mathcal{D} = \{(x_i, S_i) \mid 1 \leq i \leq m\}$, where $x_i \in \mathcal{X}$ is a $d$-dimensional feature vector to represent the $i$-th sample and $S_i \subseteq \mathcal{Y}$ is the associated candidate label set, among which only one label is correct. PLL aims to induce a multi-class classifier $f : \mathcal{X} \rightarrow \mathcal{Y}$ from $\mathcal{D}$, which is very challenging as the ground-truth label of a sample is concealed in the candidate label set.

---

*Corresponding author

37th Conference on Neural Information Processing Systems (NeurIPS 2023).

The key to solving PLL is label disambiguation, i.e., identifying the ground-truth label of a sample from its candidate label set. For example, [6, 12] considered the ground-truth label as a latent variable and identified it through an iterative refining procedure. [3] narrowed the candidate label set through a sparsity-based self-training procedure. Some works [19, 22, 23] leveraged the similarity relationship of samples in the feature space to disambiguation, i.e., samples are similar to each other in the feature space are likely to share the same ground-truth label. Recently, some researches exploited the label space to achieve label disambiguation [2, 7, 13]. For example, SDIM [2] first built a pairwise dissimilarity matrix through the candidate label sets, and then maximized the difference of the label confidence between two samples if their pairwise dissimilarity between them is large according to the constructed dissimilarity matrix. However, the dissimilarity matrix constructed by SDIM is predefined and relatively sparse, which depresses its effectiveness.

Realizing the effectiveness of the dissimilarity relationship in PLL, we propose a novel PLL method named DPCLS (partial label learning with Dissimilarity Propagation guided Candidate Label Shrinkage). Specifically, we first construct a semantic dissimilarity matrix by considering the complement of the intersection of the candidate label set, i.e., if two samples do not share any common candidate labels, they must belong to different classes, and their semantic dissimilarity is large. The above constructed dissimilarity relationships are still sparse and fixed, we therefore propose to propagate the initial dissimilarity relationships to the whole data set by leveraging the local geometric structure of samples, i.e., if two samples are similar to each other in feature space, they are expected to share the similar dissimilarity codings. Second, to include the enhanced semantic dissimilarity matrix in label disambiguation, we design a second-order similarity matrix by multiplying the label confidence matrix with its transpose. Under the ideal condition, the enhanced semantic dissimilarity matrix and the similarity matrix naturally form an adversarial relationship, i.e., a larger (resp. smaller) dissimilarity between two samples means a smaller (resp. larger) similarity between them. By exploiting this adversarial prior, the enhanced dissimilarity matrix can shrink the solution space of the label confidence matrix, and meanwhile, the similarity matrix induced from the label confidence matrix also contributes to build a better dissimilarity matrix. We theoretically confirm the above statement under some general conditions. Furthermore, we extend our method to a kernel version to model the non-linear structure of samples. The proposed model is finally formulated as a constrained regression problem with adversarial learning and graph regularization, which is optimized by inexact augmented Lagrange multiplier (IALM). Extensive experiments on artificial and real-world partial label data sets demonstrate the effectiveness of the proposed PLL method.

## 2   The Proposed Method

**Basic Model**
Let $\mathbf{X} = [x_1, x_2, \cdots, x_m]^\mathsf{T} \in \mathbb{R}^{m \times d}$ denote the feature matrix, where $m$ and $d$ represent the number of samples and the dimension of features. $\mathbf{Y} = [y_1, y_2, \cdots, y_m]^\mathsf{T} \in \{0, 1\}^{m \times q}$ represents the partial label matrix, where $q$ is the number of classes. Moreover, $y_{ij} = 1$ indicates that the $j$-th label is one of the candidate labels of the sample $x_i$. Note that in the candidate label set, only one label is correct.

To fulfill PLL, we first build the following constrained regression model

$$\min_{\mathbf{W}, \mathbf{F}} \|\mathbf{XW} - \mathbf{F}\|_F^2 + \lambda \|\mathbf{W}\|_F^2$$
$$\text{s.t. } \mathbf{F1}_q = \mathbf{1}_m, \mathbf{0}_{m \times q} \leq \mathbf{F} \leq \mathbf{Y}, \tag{1}$$

where $\mathbf{F} \in \mathbb{R}^{m \times q}$ is the label confidence matrix with $\mathbf{F}_{ij}$ representing the probability of the $j$-th label being the ground-truth label for the $i$-th sample. $\mathbf{W} \in \mathbb{R}^{d \times q}$ is the coefficient matrix that maps the feature matrix to the label confidence matrix $\mathbf{F}$. To avoid over-fitting, we impose the widely-used squared Frobenius norm on $\mathbf{W}$ as the regularization term, which is introduced by $\lambda > 0$. $\mathbf{1}_q \in \mathbb{R}^q$ and $\mathbf{1}_m \in \mathbb{R}^m$ are two all ones vectors, $\mathbf{0}_{m \times q} \in \mathbb{R}^{m \times q}$ is an all zeroes matrix. The first constraint $\mathbf{F1}_q = \mathbf{1}_m$ normalizes the label confidence vector for all samples. The second constraint $\mathbf{0}_{m \times q} \leq \mathbf{F} \leq \mathbf{Y}$ means each element of $\mathbf{F}$ is no less than 0, and no more than the corresponding element of $\mathbf{Y}$, which implies the ground-truth label of each sample must reside in the candidate label set and the label confidence of the non-candidate labels must be 0. We initialize the label confidence matrix as $\mathbf{F}_{ij} = 1/\sum_j y_{ij}$ if $y_{ij}$=1, otherwise $\mathbf{F}_{ij} = 0$.

By minimizing Eq. (1), we construct a linear regression model that maps the feature space to the label confidence $\mathbf{F}$. We assume that the mapping from the features to the ground-truth label may

be easier, while that to the false-positive label residing in the candidate label set is relatively harder. Accordingly, optimizing Eq. (1) will help disambiguate the candidate labels and produce a preliminary label confidence matrix by exploring the useful information in the feature space.

**Dissimilarity Propagation guided Candidate Label Shrinkage**

To further exploit the valuable information in the label space, we first use candidate labels to construct a dissimilarity matrix $\mathbf{D}_0 \in \mathbb{R}^{m \times m}$, i.e.,

$$\mathbf{D}_{0ij} = \begin{cases} 1, & \text{if } y_i y_j^\mathsf{T} = 0 \\ 0, & \text{otherwise.} \end{cases} \tag{2}$$

If $y_i y_j^\mathsf{T} = 0$, the $i$-th sample $x_i$ and the $j$-th sample $x_j$ do not share any common candidate labels, which means they must belong to the different classes. Otherwise, $x_i$ and $x_j$ have a chance to belong to the same class. Therefore, $\mathbf{D}_0$ indicates the semantic dissimilarity of samples. We then multiply the label confidence matrix with its transpose to create a similarity matrix termed as $\mathbf{F}\mathbf{F}^\mathsf{T}$, whose $(i, j)$-th element indicates the similarity between $x_i$ and $x_j$. *As the semantic dissimilarity matrix $\mathbf{D}_0$ and similarity matrix $\mathbf{F}\mathbf{F}^\mathsf{T}$ form an adversarial relationship,* i.e., a larger (resp. smaller) element in $\mathbf{D}_0$ implies a smaller (resp. larger) element in $\mathbf{F}\mathbf{F}^\mathsf{T}$, we use this adversarial prior to shrink the solution space of $\mathbf{F}$ by:

$$\min_{\mathbf{F}} \left\| \mathbf{D}_0 \odot \mathbf{F}\mathbf{F}^\mathsf{T} \right\|_1, \tag{3}$$

where $\odot$ denotes the element-wise product of two matrices and $||\cdot||_1$ represents the $l_1$ norm (i.e., the sum of absolute values of all elements in a matrix). Minimizing Eq. (3) ensues that the positive elements of $\mathbf{D}_0$ and $\mathbf{F}\mathbf{F}^\mathsf{T}$ will lie in the different locations. Unfortunately, directly minimizing Eq. (3) cannot help produce a better label confidence matrix $\mathbf{F}$, because $\mathbf{D}_0$ is inferred from the candidate label set, and the $(i, j)$-th element of $\mathbf{D}_0$ is positive only when $y_i y_j^\mathsf{T} = 0$, while $\mathbf{F}$ is upper bounded by $\mathbf{Y}$, the $(i, j)$-th element of $\mathbf{F}\mathbf{F}^\mathsf{T}$ is positive only when $y_i y_j^\mathsf{T} \neq 0$. That is the locations of the positive elements of $\mathbf{D}_0$ and $\mathbf{F}\mathbf{F}^\mathsf{T}$ are complementary.

To solve this problem, we propose to promote the initial dissimilarity matrix $\mathbf{D}_0$ to produce a denser dissimilarity matrix $\mathbf{D} \in \mathbb{R}^{m \times m}$ by a novel dissimilarity propagation method. Specifically, we leverage the local geometric structure of samples to enhance $\mathbf{D}_0$. Note that each column of $\mathbf{D}$ (e.g., $\mathbf{D}_{.i}$) can represent the dissimilarity relationships between a sample (e.g., $x_i$) and the other samples. If two samples $x_i$ and $x_j$ are close to each other in the feature space, their dissimilarity relationships ($\mathbf{D}_{.i}$ and $\mathbf{D}_{.j}$) should also be similar. To capture the feature similarity, we build a local geometric matrix $\mathbf{S} \in \mathbb{R}^{m \times m}$ using a radial basis function (RBF) kernel:

$$\mathbf{S}_{ij} = \begin{cases} \exp(-||x_i - x_j||_2^2 / \sigma^2), & \text{if } j \in \mathcal{N}_i \\ 0, & \text{otherwise,} \end{cases} \tag{4}$$

where $j \in \mathcal{N}_i$ indicates that $x_j$ is a $k$-nearest neighbor of $x_i$, and $\sigma$ is a hyper-parameter controlling the bandwidth of the RBF kernel. Based on $\mathbf{S}$, the dissimilarity propagation guided candidate label shrinkage module becomes

$$\min_{\mathbf{F},\mathbf{D}} \left\| \mathbf{D} \odot \mathbf{F}\mathbf{F}^\mathsf{T} \right\|_1 + \sum_{i,j=1}^m \mathbf{S}_{ij} \left\| \mathbf{D}_{.i} - \mathbf{D}_{.j} \right\|_2^2 \tag{5}$$

$$\text{s.t. } \mathbf{0}_{m \times m} \leq \mathbf{D} \leq \mathbf{1}_{m \times m}, \mathbf{D}_{ij} = \mathbf{D}_{0ij}, \text{if } \mathbf{D}_{0ij} = 1.$$

The second term makes the highly similar samples in the feature samples share the similar dissimilarity codings. Moreover, $\mathbf{D}_{ij} = \mathbf{D}_{0ij}$ if $\mathbf{D}_{0ij} = 1$ means the reliable semantic relationship should be retained in $\mathbf{D}$. In order to make $\mathbf{D}$ a well-defined dissimilarity matrix, each element in $\mathbf{D}$ should lie in the range of [0,1]. Note that in Eq. (5), both $\mathbf{F}$ and $\mathbf{D}$ are optimization variables. By minimizing Eq. (5), the enhanced semantic dissimilarity matrix will shrink the solution space of $\mathbf{F}$ and at the same time $\mathbf{F}$ will also help promote the quality of $\mathbf{D}$ by the adversarial term.

**Model Formulation**

Taking all the above considerations into account, the proposed model finally becomes:

$$\min_{\mathbf{W},\mathbf{F},\mathbf{D}} \|\mathbf{X}\mathbf{W} - \mathbf{F}\|_F^2 + \lambda \|\mathbf{W}\|_F^2 + \alpha \left\| \mathbf{D} \odot \mathbf{F}\mathbf{F}^\mathsf{T} \right\|_1 + \beta \text{Tr}(\mathbf{D}\mathbf{L}\mathbf{D}^\mathsf{T}) \tag{6}$$

$$\text{s.t. } \mathbf{F}\mathbf{1}_q = \mathbf{1}_m, \mathbf{0}_{m \times q} \leq \mathbf{F} \leq \mathbf{Y}, \mathbf{0}_{m \times m} \leq \mathbf{D} \leq \mathbf{1}_{m \times m}, \mathbf{D}_{ij} = \mathbf{D}_{0ij}, \text{if } \mathbf{D}_{0ij} = 1,$$

where $\mathbf{L} \in \mathbb{R}^{m \times m} = \mathbf{D_S} - \mathbf{S}$ is a graph Laplacian matrix, and $\mathbf{D_S} \in \mathbb{R}^{m \times m}$ is a diagonal matrix with the $i$-th diagonal element $\mathbf{D}_{\mathbf{S}ii} = \sum_{j=1}^m \mathbf{S}_{ij}$. $\text{Tr}(\cdot)$ returns the trace of a matrix. $\alpha, \beta \geq 0$ are two hyper-parameters to balance different terms. By solving Eq. (6), the semantic dissimilarity is enhanced by a dissimilarity propagation process, and further utilized to shrink the solution space of the label confidence matrix by a novel adversarial prior.

# 3 Optimization and Setting

We adopt IALM to solve the problem in Eq. (6). To simplify Eq. (6), we introduce an auxiliary matrix $\mathbf{A} = \mathbf{D} \in \mathbb{R}^{m \times m}$ and equivalently rewrite it as

$$\min_{\mathbf{W},\mathbf{F},\mathbf{D},\mathbf{A}} \|\mathbf{XW} - \mathbf{F}\|_F^2 + \lambda \|\mathbf{W}\|_F^2 + \alpha \left\| \mathbf{A} \odot \mathbf{FF}^\mathsf{T} \right\|_1 + \beta \mathrm{Tr}(\mathbf{DLD}^\mathsf{T}) \tag{7}$$

$$\text{s.t. } \mathbf{F1}_q = \mathbf{1}_m, \mathbf{0}_{m \times q} \leq \mathbf{F} \leq \mathbf{Y}, \mathbf{D} = \mathbf{A}, \mathbf{0}_{m \times m} \leq \mathbf{A} \leq \mathbf{1}_{m \times m}, \mathbf{A}_{ij} = \mathbf{D}_{0ij}, \text{if } \mathbf{D}_{0ij} = 1.$$

The solution of Eq. (7) can be obtained by solving the following augmented Lagrange equation:

$$\min_{\mathbf{W},\mathbf{F},\mathbf{D},\mathbf{A}} \|\mathbf{XW} - \mathbf{F}\|_F^2 + \lambda \|\mathbf{W}\|_F^2 + \alpha \left\| \mathbf{A} \odot \mathbf{FF}^\mathsf{T} \right\|_1 + \beta \mathrm{Tr}(\mathbf{DLD}^\mathsf{T}) + \langle \mathbf{\Phi}, \mathbf{D} - \mathbf{A} \rangle + \frac{\mu}{2} \|\mathbf{D} - \mathbf{A}\|_F^2$$

$$\text{s.t. } \mathbf{F1}_q = \mathbf{1}_m, \mathbf{0}_{m \times q} \leq \mathbf{F} \leq \mathbf{Y}, \mathbf{0}_{m \times m} \leq \mathbf{A} \leq \mathbf{1}_{m \times m}, \mathbf{A}_{ij} = \mathbf{D}_{0ij}, \text{if } \mathbf{D}_{0ij} = 1, \tag{8}$$

where $\mathbf{\Phi} \in \mathbb{R}^{m \times m}$ is the Lagrange multiplier matrix, $\mu \geq 0$ is a penalty parameter, and $\langle \cdot, \cdot \rangle$ returns the inner product of two matrices. We can optimize Eq. (8) by solving the following subproblems alternatively and iteratively.

*1)* $\mathbf{W}$ subproblem is formulated as

$$\min_{\mathbf{W}} \|\mathbf{XW} - \mathbf{F}\|_F^2 + \lambda \|\mathbf{W}\|_F^2, \tag{9}$$

**Kernel Extension**
The linear mapping in Eq. (9) may fail to model the nonlinear relationship. Therefore, we extend the above model to a kernel-based non-linear version. Let $\phi(\cdot) : \mathbb{R}^d \to \mathbb{R}^h$ denote the feature transformation that maps the origin feature space $\mathbf{X}$ to a higher dimensional Hilbert space $\phi(\mathbf{X})$. According to the Representer Theorem, $\mathbf{W}$ can be expressed as a linear combination of the input features, i.e., $\mathbf{W} = \phi(\mathbf{X})^\mathsf{T} \mathbf{H}$, where $\mathbf{H} \in \mathbb{R}^{m \times q}$ stores the combination weights. Then, we have $\phi(\mathbf{X})\mathbf{W} = \mathbf{KH}$, where $\mathbf{K} = \phi(\mathbf{X})\phi(\mathbf{X})^\mathsf{T} \in \mathbb{R}^{m \times m}$ is the kernel matrix and each element $\mathbf{K}_{ij} = \mathcal{K}(x_i, x_j)$. Finally, the nonlinear version is represented as:

$$\min_{\mathbf{H},\mathbf{b}} \left\| \mathbf{KH} + \mathbf{1}_m \mathbf{b}^\mathsf{T} - \mathbf{F} \right\|_F^2 + \lambda \mathrm{Tr}(\mathbf{H}^\mathsf{T} \mathbf{KH}), \tag{10}$$

where $\mathbf{b} \in \mathbb{R}^q$ is the bias term. When the first derivatives of $\mathbf{H}$ and $\mathbf{b}$ reach 0, Eq. (10) is solved, i.e.,

$$\mathbf{H} = \left( \mathbf{K} + \lambda \mathbf{I}_{m \times m} - \frac{\mathbf{1}_m \mathbf{1}_m^\mathsf{T} \mathbf{K}}{m} \right)^{-1} \left( \mathbf{F} - \frac{\mathbf{1}_m \mathbf{1}_m^\mathsf{T} \mathbf{F}}{m} \right), \mathbf{b} = \frac{1}{m} \left( \mathbf{F}^\mathsf{T} \mathbf{1}_m - \mathbf{H}^\mathsf{T} \mathbf{K}^\mathsf{T} \mathbf{1}_m \right), \tag{11}$$

where $\mathbf{I}_{m \times m} \in \mathbb{R}^{m \times m}$ denotes an identity matrix. In the experiments, we use the RBF kernel as the kernel function, i.e., $\mathcal{K}(x_i, x_j) = \exp(-\|x_i - x_j\|_2^2/\sigma^2)$, for our method and the compared ones.
*2)* $\mathbf{F}$ subproblem is written as

$$\min_{\mathbf{F}} \|\mathbf{F} - \mathbf{P}\|_F^2 + \alpha \left\| \mathbf{A} \odot \mathbf{FF}^\mathsf{T} \right\|_1 \tag{12}$$

$$\text{s.t. } \mathbf{F1}_q = \mathbf{1}_m, \mathbf{0}_{m \times q} \leq \mathbf{F} \leq \mathbf{Y},$$

where $\mathbf{P} = \mathbf{KH} + \mathbf{1}_m \mathbf{b}^\mathsf{T} \in \mathbb{R}^{m \times q}$ is the output matrix of the model. Eq. (12) can be formulated as a standard quadratic programming (QP) problem, and solved by any QP tools. The detailed derivation process can be found in **Section A** of the supplementary file.
*3)* $\mathbf{D}$ subproblem is represented as

$$\min_{\mathbf{D}} \beta \mathrm{Tr}(\mathbf{DLD}^\mathsf{T}) + \frac{\mu}{2} \left\| \mathbf{D} - \mathbf{A} + \frac{\mathbf{\Phi}}{\mu} \right\|_F^2. \tag{13}$$

Eq. (13) reaches the minimum when its first-order derivative with respect to $\mathbf{D}$ vanishes, leading to

$$\mathbf{D} = (\mu \mathbf{A} - \mathbf{\Phi})(2\beta \mathbf{L} + \mu \mathbf{I}_{m \times m})^{-1}. \tag{14}$$

*4)* $\mathbf{A}$ subproblem is expressed as

$$\min_{\mathbf{A}} \alpha \left\| \mathbf{A} \odot \mathbf{FF}^\mathsf{T} \right\|_1 + \frac{\mu}{2} \left\| \mathbf{D} - \mathbf{A} + \frac{\mathbf{\Phi}}{\mu} \right\|_F^2 \tag{15}$$

$$\text{s.t. } \mathbf{0}_{m \times m} \leq \mathbf{A} \leq \mathbf{1}_{m \times m}, \mathbf{A}_{ij} = \mathbf{D}_{0ij}, \text{if } \mathbf{D}_{0ij} = 1.$$

Eq. (15) can solved element-wisely, i.e.,

$$\mathbf{A} = \mathcal{T} \left( \mathcal{T}_1 \left( \mathcal{T}_0 \left( \frac{\mu \mathbf{D} + \mathbf{\Phi} - \alpha \mathbf{FF}^\mathsf{T}}{\mu} \right) \right) \right), \tag{16}$$

---
**Algorithm 1** The Pseudo Code of the Proposed Method
---
**Input**: $\mathcal{D}$: the partial label training set; $\lambda, \alpha, \beta, k$: the parameters of model; $\widehat{x}$: an unseen test sample
**Output**: $\widehat{y}$: the predicted label for sample $\widehat{x}$
 1: Construct the dissimilarity matrix $\mathbf{D}_0$ according to Eq. (2) and the kernel matrix $\mathbf{K} = [\mathcal{K}(x_i, x_j)]_{m \times m}$
 2: Initialize $\mathbf{D} = \mathbf{A} = \mathbf{\Phi} = \mathbf{0}_{m \times m}, \mu = 10^{-4}$
 3: **while** not converged **do**
 4:    Update $\mathbf{H}$ and $\mathbf{b}$ by Eq. (11)
 5:    Update $\mathbf{F}$ by solving Eq. (12)
 6:    Update $\mathbf{D}$ by Eq. (14)
 7:    Update $\mathbf{A}$ by Eq. (16)
 8:    Update $\mathbf{\Phi}, \mu$ by Eq. (17)
 9:    Check the convergence condition $\|\mathbf{D} - \mathbf{A}\|_\infty < 10^{-8}$
10: **end while**
11: Return the predicted label $\widehat{y}$ according to Eq. (18).
---

where $\mathcal{T}, \mathcal{T}_1, \mathcal{T}_0$ are three thresholding operators in elementwise, i.e., $\mathcal{T}(\mathbf{C}_{ij}) = 1,$ if $\mathbf{D}_{0ij} = 1,$ $\mathcal{T}_1(\mathbf{C}_{ij}) := \min(1, \mathbf{C}_{ij}), \mathcal{T}_0(\mathbf{C}_{ij}) := \max(0, \mathbf{C}_{ij}).$

Finally, the Lagrangian multiplier matrix and $\mu$ are updated by

$$\begin{cases} \mathbf{\Phi} \leftarrow \mathbf{\Phi} + \mu(\mathbf{D} - \mathbf{A}) \\ \mu \leftarrow \min(1.1\mu, \mu_{\max}), \end{cases} \tag{17}$$

where $\mu_{\max}=10^{10}$ is a predefined upper bound for $\mu$.
**Model Prediction**
Given an unseen test example $\widehat{x}$, our method predicts its label by

$$\widehat{y} = \arg \max_k \sum_{i=1}^m \mathbf{H}_{ik} \mathcal{K}(\widehat{x}, x_i) + \mathbf{b}_k. \tag{18}$$

The overall pseudo code of our method is summarized in Algorithm 1, where it stops when $\|\mathbf{D} - \mathbf{A}\|_\infty < 10^{-8}$, where $\| \cdot \|_\infty$ denotes the infinity norm of a matrix.
**Hyper-parameter Settings of Our Method**
Parameter $\lambda$ is used to control the model complexity. A too large (resp. small) value of $\lambda$ will lead to under-fitting (resp. over-fitting). Therefore, we set $\lambda=0.05$ for our method. Parameters $\alpha$ and $\beta$ are used to control the importance of the adversarial term and dissimilarity propagation term respectively. According to a number of experiments, we fix $\beta = 0.001$ and select $\alpha$ from $\{0.001, 0.01\}$. Parameter $k$ controls the number of $k$-nearest neighbors. Following the previous works [16, 22], we set $k=10$.
**Computational Complexity Analysis**
The computational complexity of Algorithm 1 is mainly determined by steps 4-7. Specifically, steps 4 and 6 involve the inversion of $m \times m$ matrices with the complexity of $\mathcal{O}(m^3)$. Step 5 solves a QP problem, leading to the complexity of $\mathcal{O}(m^3q^3)$. Step 7 can be efficiently solved by linear thresholding operations with the complexity of $\mathcal{O}(m^2)$. Therefore, the overall computational complexity of Algorithm 1 in each iteration is $\mathcal{O}(2m^3 + m^2 + m^3q^3)$. More analysis about computational complexity can be found in **Section A** of the supplementary file.

## 4   Theoretical Analysis

**Theorem 1.** *The square loss function $\ell$ of DPCLS can be rewritten as $\|\mathbf{F_G} + \mathbf{N} - \mathbf{XW}\|_F^2$, where $\mathbf{F_G} \in \mathbb{R}^{m \times q}$ and $\mathbf{N} \in \mathbb{R}^{m \times q}$ are ground-truth label matrix and false-positive label matrix respectively. Let $\mathcal{H} = \mathbf{W} \times \mathbf{N}$ represent the family of functions for DPCLS, where the linear function $(\mathbf{W}, \mathbf{N}) \in \mathcal{H}$. Suppose the complexity of $\mathbf{W}$ and the sparsity of $\mathbf{N}$ are upper bounded by $\epsilon_1$ and $\epsilon_2$ respectively, i.e., $\|\mathbf{W}\|_F \le \epsilon_1$ and $\|\mathbf{N}\|_1 \le \epsilon_2$. The Rademacher complexity of DPCLS with square loss $\ell$ is upper bounded as follow*

$$\hat{\mathcal{R}}_S(\ell \circ \mathcal{H}) \le \frac{2\sqrt{2}q(\sqrt{mq}\epsilon_1 + \epsilon_2)}{m}. \tag{19}$$

**Lemma 1.** *[11] Denote $\mathcal{H}$ be a family of functions and $S = \{x_1, x_2, ..., x_m\}$ is a set of fixed samples. Loss function $\ell$ is upper bounded by $\Theta \ge 0$, then for any $\delta > 0$, with probability at least $1 - \delta$, for all $h \in \mathcal{H}$ we have*

$$\mathcal{L}(h) \le \mathcal{L}_S(h) + \hat{\mathcal{R}}_S(\ell \circ \mathcal{H}) + 3\Theta \sqrt{\frac{log(2/\delta)}{2m}}, \tag{20}$$

*where $\mathcal{L}(h)$ and $\mathcal{L}_\mathcal{S}(h)$ are generalization error and empirical error to $h$ respectively.*

The detailed proof of **Theorem 1** is given in the **Section B** of the supplementary file. According to **Lemma 1** and **Theorem 1**, we have

$$\mathcal{L}(h) \leq \mathcal{L}_\mathcal{S}(h) + \frac{2\sqrt{2}q(\sqrt{mq}\epsilon_1 + \epsilon_2)}{m} + 3\Theta\sqrt{\frac{log(2/\delta)}{2m}}. \tag{21}$$

The Rademacher complexity is bounded by the sum of the complexity of classifier $\mathbf{W}$ and the sparsity of false-positive label matrix $\mathbf{N}$. When the number of false-positive labels is small, which leads to a better generalization performance. Moreover, more training samples (a larger $m$) will also promote the generalization performance.

**Theorem 2.** *Denote $\mathbf{F} \in \{0,1\}^{m \times q}$ and $\mathbf{D} \in \{0,1\}^{m \times m}$ the partial label matrix and the to be optimized semantic dissimilarity matrix. Let $\mathbf{F_G}$ and $\hat{\mathbf{D}}$ be the ground-truth label matrix and the ground-truth dissimilarity matrix. Suppose the smallest eigenvalue of $\hat{\mathbf{D}}$ and $\mathbf{L}$ are $\lambda_{\hat{\mathbf{D}}}$ and $\lambda_\mathbf{L}$ respectively ($\lambda_{\hat{\mathbf{D}}} \geq 0, \lambda_\mathbf{L} \geq 0$). Let $\left\|\bar{\mathbf{\Delta}}_\mathbf{F}\right\|_F$ be the average distance of each sample between $\mathbf{F_G}$ and $\mathbf{F}$ (i.e., $\left\|\bar{\mathbf{\Delta}}_\mathbf{F}\right\|_F = \frac{1}{m}\left\|\mathbf{F_G} - \mathbf{F}\right\|_F$) and $\left\|\bar{\mathbf{\Delta}}_\mathbf{D}\right\|_F$ be the average distance of each corresponding position between $\hat{\mathbf{D}}$ and $\mathbf{D}$ (i.e., $\left\|\bar{\mathbf{\Delta}}_\mathbf{D}\right\|_F = \frac{1}{m^2}\left\|\hat{\mathbf{D}} - \mathbf{D}\right\|_F$). Then we have*

$$\begin{aligned}
\left\|\bar{\mathbf{\Delta}}_\mathbf{F}\right\|_F &\leq \frac{q}{\lambda_{\hat{\mathbf{D}}}}\left\|\mathbf{D} - \hat{\mathbf{D}}\right\|_F + \frac{2\sqrt{q}}{\lambda_{\hat{\mathbf{D}}}\sqrt{m}}\left\|\hat{\mathbf{D}}\right\|_F, \\
\left\|\bar{\mathbf{\Delta}}_\mathbf{D}\right\|_F &\leq \frac{1}{\lambda_\mathbf{L}m}\left\|\mathbf{FF}^\mathsf{T} - \mathbf{F_G}\mathbf{F_G}^\mathsf{T}\right\|_F + \frac{2}{\lambda_\mathbf{L}m}\left\|\mathbf{L}\right\|_F + \frac{1}{\lambda_\mathbf{L}m}.
\end{aligned} \tag{22}$$

The proof can be found in **Section B** of the supplementary file. From **Theorem 2**, we can find that as the number of samples $m$ increases, the upper bound of $\left\|\bar{\mathbf{\Delta}}_\mathbf{F}\right\|_F$ decreases, which indicates that more training samples will push the partial label matrix to be close to the ground-truth one and achieve better PLL performance. Moreover, a smaller error between $\mathbf{D}$ and $\hat{\mathbf{D}}$ implies a smaller upper bound of $\left\|\bar{\mathbf{\Delta}}_\mathbf{F}\right\|_F$, which indicates a better dissimilarity matrix can help achieve a better label matrix. Similarly, a larger number of training samples will reduce the distance between $\mathbf{D}$ and $\hat{\mathbf{D}}$, and a smaller error between $\mathbf{F}$ and $\mathbf{F_G}$ implies a smaller upper bound of $\left\|\bar{\mathbf{\Delta}}_\mathbf{D}\right\|_F$, suggesting a better dissimilarity matrix. *As a summary, we prove that, under some general assumptions, a better dissimilarity matrix will help produce a better label matrix, and vice versa. Therefore, the rationality of the proposed adversarial prior is theoretically proved.*

## 5 Experiment and Analysis

To demonstrate the effectiveness of the proposed model, we compared DPCLS with eight shallow PLL algorithms, which were configured by the suggested parameters in the literature, i.e., CLPL [1], PL-SVM [12], PL-KNN [5], PL-DA [18], IPAL [22], AGGD [16], PL-CLA [13], SDIM [2]. Those methods were evaluated on 10 synthetic data sets and 7 real-world data sets, whose details can be found in **Section C** of the supplementary file. Three deep learning PLL methods RC [4], PRODEN [10], and CAVL [21] were also evaluated as comparison on the real-world data sets. Ten runs of $50\%/50\%$ random train/test splits were performed on each data set, and the average classification accuracy and the standard deviation were recorded.

### 5.1 Performance on Synthetic Data Sets

Following the widely-used partial label data generation protocol [1], 10 synthetic data sets were used to generate artificial partial label data sets. Specifically, three parameters $r, p, \epsilon$ control the generation process, i.e., $p$ controls the proportion of partial label examples, $r$ controls the number of false-positive labels, and $\epsilon$ controls the probability of a specific false-positive label co-occurs with the ground-truth label.

Fig. 1 illustrates the classification accuracy of six data sets (Glass, Steel, Ecoli, Yeast, Optdigits and Usps) as the co-occurring probability $\epsilon$ varies from 0.1 to 0.7 with step-size 0.1 ($p$=1, $r$=1). In general, the proposed model clearly exceeds the compared methods with different $\epsilon$, and achieves the highest accuracy in 33 out of 36 cases. Moreover, as $\epsilon$ increases, it is more difficult to distinguish the

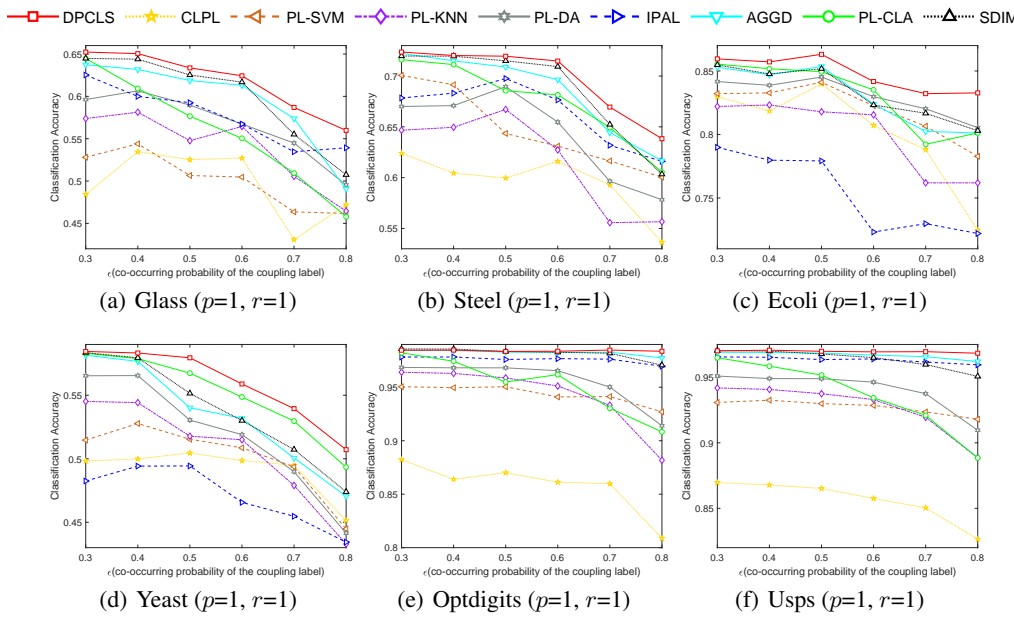

Figure 1: The classification accuracy of each algorithm as $\epsilon$ increases from 0.1 to 0.7 with $p$=1, $r$=1.

Table 1: Classification accuracy (mean±std) of each comparing algorithm on four synthetic data sets as $r$ increases from 3 to 6. ●/○ indicates whether the accuracy of DPCLS is statistically superior/inferior to the compared algorithm according to the pairwise t-test at 0.05 significance level.

| Data set | r | DPCLS | CLPL | PL-SVM | PL-KNN | PL-DA | IPAL | AGGD | PL-CLA | SDIM |
|---|---|---|---|---|---|---|---|---|---|---|
| Isolet | 3 | .912±.008 | .604±.023● | .668±.034● | .695±.025● | .782±.020● | .793±.018● | .898±.011● | .880±.007● | **.917±.007** |
| | 4 | **.899±.011** | .594±.023● | .577±.057● | .661±.026● | .768±.022● | .772±.017● | .880±.013● | .855±.011● | .891±.015 |
| | 5 | **.880±.012** | .535±.015● | .543±.070● | .668±.018● | .751±.016● | .790±.010● | .855±.017● | .845±.018● | .849±.017● |
| | 6 | **.858±.017** | .525±.020● | .452±.062● | .645±.020● | .733±.020● | .776±.011● | .824±.015● | .824±.015● | .824±.018● |
| Orl | 3 | **.807±.040** | .780±.037 | .262±.034● | .247±.033● | .347±.061● | .715±.064● | .764±.054 | .735±.055● | .744±.044● |
| | 4 | **.776±.049** | .743±.031 | .196±.052● | .216±.030● | .287±.042● | .665±.065● | .730±.065 | .673±.066● | .692±.067● |
| | 5 | **.736±.041** | .704±.032 | .158±.029● | .189±.027● | .247±.040● | .594±.045● | .677±.067● | .607±.049● | .622±.056● |
| | 6 | **.669±.042** | .660±.045 | .114±.024● | .159±.025● | .218±.029● | .536±.047● | .646±.073 | .545±.050● | .556±.053● |
| Amazon | 3 | **.062±.007** | .058±.007 | .041±.006● | .021±.004● | .022±.004● | .060±.009 | .056±.010 | .055±.008● | .047±.006● |
| | 4 | **.057±.007** | .055±.007 | .043±.005● | .020±.005● | .021±.005● | .055±.009 | .054±.009 | .053±.007 | .045±.005● |
| | 5 | **.055±.009** | .050±.007 | .043±.005● | .020±.005● | .022±.005● | .050±.009 | .047±.008● | .044±.009● | .037±.007● |
| | 6 | **.048±.009** | .042±.007 | .029±.007● | .020±.003● | .020±.003● | .046±.009 | .043±.008 | .041±.006● | .032±.006● |
| Bookmark | 3 | **.337±.007** | .248±.007● | .208±.013● | .115±.017● | .178±.014● | .286±.012● | .333±.009 | .291±.009● | .305±.010● |
| | 4 | **.326±.008** | .244±.009● | .190±.020● | .121±.028● | .170±.013● | .282±.010● | .325±.007 | .284±.011● | .300±.011● |
| | 5 | **.323±.009** | .240±.010● | .169±.019● | .116±.025● | .189±.051● | .278±.009● | .319±.005 | .278±.010● | .298±.011● |
| | 6 | **.322±.009** | .240±.010● | .148±.013● | .103±.029● | .186±.043● | .270±.009● | .316±.007 | .276±.009● | .295±.007● |

false-positive label and the ground-truth label, and PLL becomes harder. Accordingly, the accuracies of all the PLL algorithms decrease, but the performance advantage of the proposed method over the compared ones becomes more salient such as on Glass and Ecoli, which proves that our method is skilled in more challenging PLL tasks.

Table 1 reports the classification accuracies of different algorithms as $r$ varies from 3 to 6 on four data sets (Isolet, Orl, Amazon, Bookmark) with more than 20 classes. Our method ranks first in 11 out of 12 cases, and ranks second in the remaining case. The improvements brought by our method are significant. For example, on Isolet with $r = 6$, our method improves the classification accuracy of the best comparison from 0.824 to 0.858. According to the pairwise t-test, our method significantly surpasses the compared ones in 103 out of 128 cases. Moreover, different methods have different characteristics and may fit different data sets. For example, CLPL performs satisfactorily on Orl and Amazon, but badly on Isolet and Bookmark; AGGD is good at Bookmark, but not at Orl.

Table 2: Classification accuracy (mean±std) of each algorithm on real-world partial label data sets. ●/○ indicates whether the accuracy of DPCLS is statistically superior/inferior to the compared algorithm according to the pairwise t-test at 0.05 significance level. "S" and "D" indicate shallow and deep PLL methods respectively.

| Type | Method | FG-NET | FG-NET3 | FG-NET5 | Lost | MSRCv2 | BirdSong | Malagasy | Soccer Player | Yahoo! News |
|------|--------|--------|---------|---------|------|--------|----------|----------|---------------|-------------|
|      | DPCLS | **.077±.009** | **.436±.017** | **.586±.011** | **.770±.024** | **.557±.014** | **.751±.009** | **.676±.004** | **.532±.002** | .626±.003 |
|      | CLPL | .058±.009● | .383±.016● | .538±.017● | .665±.019● | .371±.010● | .610±.012● | .675±.016 | .497±.002● | .544±.004● |
|      | PL-SVM | .052±.010● | .357±.022● | .511±.026● | .578±.078● | .310±.060● | .682±.023● | .564±.061● | .500±.002● | .546±.006● |
|      | PL-KNN | .038±.005● | .287±.022● | .433±.019● | .334±.021● | .391±.023● | .657±.014● | .573±.007● | .493±.002● | .383±.003● |
| S    | PL-DA | .042±.004● | .166±.050● | .255±.070● | .309±.069● | .416±.022● | .690±.013● | .606±.008● | .495±.003● | .397±.004● |
|      | IPAL | .052±.006● | .347±.015● | .510±.016● | .610±.020● | .494±.024● | .722±.006● | .621±.017● | .530±.005 | .618±.007● |
|      | AGGD | .075±.010 | .423±.016 | .568±.018● | .702±.024● | .477±.019● | .722±.014● | .593±.050● | .527±.003● | .616±.004● |
|      | PL-CLA | .074±.011 | .424±.020 | .571±.015● | .696±.021● | .470±.016● | .722±.012● | .654±.005● | .525±.003● | .606±.004● |
|      | SDIM | .073±.009 | .423±.022 | .568±.019● | .736±.023● | .475±.016● | .724±.012● | .643±.007● | .524±.003● | .607±.004● |
|      | RC | .072±.009 | .391±.012● | .488±.020● | .740±.026● | .446±.019● | .715±.007● | .664±.004● | **.532±.004** | .620±.003● |
| D    | PRODEN | .071±.009 | .415±.016● | .567±.025● | .712±.032● | .430±.019● | .704±.013● | .665±.017● | .528±.004● | .620±.003● |
|      | CAVL | .071±.006 | .365±.020● | .488±.021● | .747±.060● | .444±.013● | .695±.017● | .668±.039 | .510±.004● | **.628±.004** |

Table 3: Win/tie/loss counts on the classification performance of DPCLS against each comparing algorithm on all data sets. D-PLL indicates the summary of deep learning PLL methods RC, PRODEN and CAVL. (I), (II) indicate the summaries on synthetic data sets and on real-world data sets respectively. "Total" denotes the summary on all the data sets.

|       | CLPL | PL-SVM | PL-KNN | PL-DA | IPAL | AGGD | PL-CLA | SDIM | D-PLL |
|-------|------|--------|--------|-------|------|------|--------|------|-------|
| (I)   | 44/8/0 | 51/1/0 | 52/0/0 | 47/5/0 | 46/6/0 | 23/29/0 | 38/14/0 | 30/22/0 | – |
| (II)  | 8/1/0 | 9/0/0 | 9/0/0 | 9/0/0 | 8/1/0 | 7/2/0 | 7/2/0 | 7/2/0 | 21/6/0 |
| Total | 52/9/0 | 60/1/0 | 61/0/0 | 56/5/0 | 54/7/0 | 30/31/0 | 45/16/0 | 37/24/0 | 21/6/0 |

Different from them, the proposed method produces excellent performance on all the evaluated data sets, suggesting its robustness to different data sets.

Table 3 (row (I)) reports win/tie/loss counts between DPCLS and eight comparing algorithms on the synthetic data sets according to the pairwise t-test at the significance level of 0.05, where we can find that DPCLS statistically outperforms other algorithms in 79.6% cases (331 out of 416) and none of the algorithms can beat DPCLS significantly.

## 5.2 Performance on Real-world Data Sets

Real-world data sets were collected from various tasks and domains. As the average size of the candidate label set of FG-NET is large, which will cause low classification accuracy on the test set. Following [2, 16] we employed the mean absolute error (MAE) to further calculate two extra evaluation indicators MAE3 and MAE5 on FG-NET, i.e., the test examples are considered to be correctly classified if the difference between the predicted age and the ground-truth age is no more than 3/5 years.

Table 2 demonstrates the classification accuracies of different methods on the real-world data sets. It is obvious that our method ranks first in all cases when compared with shallow PLL algorithms. Moreover, according to the pairwise t-test, our method statistically outperforms PL-SVM, PL-KNN and PL-DA on all real-world data sets and statistically surpasses AGGD, PL-CLA and SDIM on six data sets except FG-NET. Furthermore, compared with the best comparisons, our method improves the classification accuracy from 0.494 to 0.557 on MSRCv2 and from 0.747 to 0.770 on Lost.

We also compare the classification accuracies of our method with three deep learning PLL methods RC, PRODEN, CAVL on the real-world data sets. Our method ranks first on six the data sets except Yahoo! News. Specifically, according to the pairwise t-test, our method statistically outperforms RC, PRODEN and CAVL on three real-world data sets (Lost, MSRCv2 and BirdSong). Furthermore, compared with the best comparisons of deep leraning PLL methods, our method improves the classification accuracy from 0.446 to 0.557 on MSRCv2 and from 0.715 to 0.751 on BirdSong. Therefore, we can conclude that our method shows competitive performance compared with the deep learning PLL methods.

Table 4: Ablation study of our method on the real-world partial label data sets.

|  | FG-NET | Lost | MSRCv2 | BirdSong | Malagasy | Soccer Player | Yahoo! News |
|---|---|---|---|---|---|---|---|
| DPCLS | **.077**±**.009** | **.770**±**.024** | **.557**±**.014** | **.751**±**.009** | **.676**±**.004** | **.532**±**.002** | **.626**±**.003** |
| DPCLS-LM | .067±.009● | .652±.023● | .357±.009● | .577±.012● | .587±.014● | .492±.002● | .447±.004● |
| DPCLS-KE | .068±.009● | .701±.023● | .388±.014● | .595±.014● | .674±.009 | .495±.002● | .463±.004● |
| DPCLS-DP | .073±.010 | .687±.027● | .466±.018● | .721±.014● | .612±.011● | .524±.003● | .604±.004● |

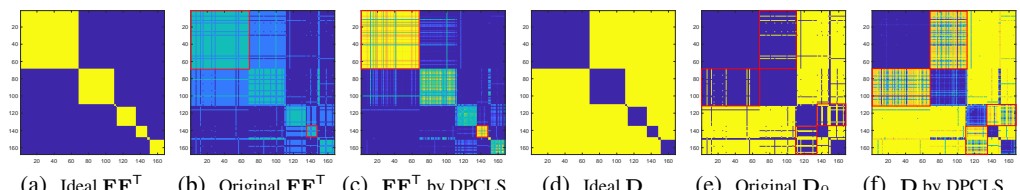

(a) Ideal $\mathbf{FF}^{\mathsf{T}}$   (b) Original $\mathbf{FF}^{\mathsf{T}}$   (c) $\mathbf{FF}^{\mathsf{T}}$ by DPCLS   (d) Ideal $\mathbf{D}$   (e) Original $\mathbf{D}_0$   (f) $\mathbf{D}$ by DPCLS

Figure 2: Visual comparison of the similarity matrix and the semantic dissimilarity matrix on data set Ecoli ($p$=1, $r$=1, $\epsilon$=0.8). (a), (d) Ideal similarity matrix and semantic dissimilarity matrix; (b), (e) Initial similarity matrix $\mathbf{FF}^{\mathsf{T}}$ and semantic dissimilarity matrix $\mathbf{D}_0$; (c), (f) The similarity matrix $\mathbf{FF}^{\mathsf{T}}$ and the dissimilarity matrix $\mathbf{D}$ produced by DPCLS. The brighter color indicates a larger value.

Table 3 (row (II)) presents the win/tie/loss counts of the proposed method against the compared ones on real-world data sets according to the pairwise t-test at significance level of 0.05. On all the real-world data sets, DPCLS achieves significantly superior performance in 85.9% cases, and gets comparable performance in the remaining cases against the comparing algorithms. Taking both the synthetic data sets and real-world data sets into account, our method accomplishes significantly better performance in 80.8% cases, which indicates the excellent classification ability of our method.

### 5.3 Further Analysis

**Visualization** Fig. 2 visually compares the ideal, the original and the enhanced similarity matrix and semantic dissimilarity matrices on data set Ecoli ($p$=1, $r$=1, $\epsilon$=0.8). Compared with the ideal similarity matrix, the value of each element in the original similarity matrix $\mathbf{FF}^{\mathsf{T}}$ is relatively small with many incorrect connections. Compared with the ideal dissimilarity matrix, the original dissimilarity matrix $\mathbf{D}_0$ is relatively sparse, and many positive elements are missed in $\mathbf{D}_0$. On the contrary, the similarity matrix and the dissimilarity matrix produced by DPCLS become denser, which are quite close to the ideal ones. Especially, when checking the areas highlighted by the red rectangle boxes, DPCLS can effectively recover the similarity relationship between samples in $\mathbf{FF}^{\mathsf{T}}$, and many positive elements missed by $\mathbf{D}_0$ have been recovered by our method. The visual comparison illustrates that our method can produce much higher-quality similarity and dissimilarity matrices, which is useful to find a high-quality label confidence matrix $\mathbf{F}$ and promote the performance of PLL.

**Ablation Study** In Table 4, we conduct an ablation study on the real-world data sets to check the necessity of the involved terms of our method. Specifically, DPCLS-LM denotes DPCLS without Dissimilarity Propagation and Kernel Extension, and DPCLS-KE and DPCLS-DP indicate DPCLS without Kernel Extension and without Dissimilarity Propagation respectively. From Table 4, we can find that both the kernel extension and the dissimilarity propagation are helpful in improving classification accuracy and taking both of them into account is the best choice.

## 6 Conclusion

In this paper, we have presented a novel PLL method. Specifically, we first construct a similarity matrix based on the multiplication of the label confidence matrix and its transpose. Then, we develop a dissimilarity matrix by exploiting the label space, and further utilize the local geometric structure of the samples to enhance the dissimilarity matrix, i.e., propagating the initial semantic dissimilarity relationships to the whole data set. The similarity and dissimilarity matrices form an adversarial relation, and the proposed model takes advantage of this novel adversarial prior to shrink the solution space of the label confidence matrix, which contributes to find the correct label in the candidate label set. Extensive experiments and comparisons on artificial and real-world partial label data sets have validated the effectiveness of our approach.

**Acknowledgments**

This work was supported in part by the National Natural Science Foundation of China under Grant 62106044 and 62072100, in part by the Natural Science Foundation of Jiangsu Province under Grant BK20210221, and in part by the ZhiShan Youth Scholar Program from Southeast University under Grant 2242022R40015.

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
