# A Solution of F subproblem

F subproblem is written as

$$\min_{\mathbf{F}} \|\mathbf{F} - \mathbf{P}\|_F^2 + \alpha \left\| \mathbf{A} \odot \mathbf{F}\mathbf{F}^{\mathsf{T}} \right\|_1 \tag{1}$$
$$\text{s.t. } \mathbf{F}\mathbf{1}_q = \mathbf{1}_m, \mathbf{0}_{m\times q} \leq \mathbf{F} \leq \mathbf{Y},$$

where $\mathbf{P} = \mathbf{KH} + \mathbf{1}_m \mathbf{b}^{\mathsf{T}} \in \mathbb{R}^{m\times q}$ is the output matrix of the model. We first reformulate Eq. (1) column-wisely:

$$\min_{\mathbf{F}} \alpha \sum_{i=1}^{q} \mathbf{F}_{.i}^{\mathsf{T}} \mathbf{A}\mathbf{F}_{.i} + \sum_{i=1}^{q} \mathbf{F}_{.i}^{\mathsf{T}} \mathbf{F}_{.i} - 2\sum_{i=1}^{q} \mathbf{P}_{.i}^{\mathsf{T}} \mathbf{F}_{.i} \tag{2}$$
$$\text{s.t. } \mathbf{F}\mathbf{1}_q = \mathbf{1}_m, \mathbf{0}_{m\times q} \leq \mathbf{F} \leq \mathbf{Y},$$

where $\mathbf{F}_{.i}$ and $\mathbf{P}_{.i}$ are the $i$-th column of $\mathbf{F}$ and $\mathbf{P}$ respectively. Denote $\mathbf{f} = vec(\mathbf{F}) \in \mathbb{R}^{mq}$, $\mathbf{p} = vec(\mathbf{P}) \in \mathbb{R}^{mq}$, $\mathbf{y} = vec(\mathbf{Y}) \in \mathbb{R}^{mq}$, where $vec(\cdot)$ is the vectorization operator. Eq. (2) can be further formulated as

$$\min_{\mathbf{f}} \frac{1}{2}\mathbf{f}^{\mathsf{T}} \left( 2\boldsymbol{\Lambda} + \frac{2}{\alpha}\mathbf{I}_{mq\times mq} \right) \mathbf{f} - \frac{2}{\alpha}\mathbf{p}^{\mathsf{T}}\mathbf{f}$$
$$\text{s.t. } \sum_{\substack{j=1,\\ j\%m=i}}^{mq} \mathbf{f}_j = 1 (\forall\, 0 \leq i \leq m-1), \mathbf{0}_{mq} \leq \mathbf{f} \leq \mathbf{y}, \tag{3}$$

where % denotes the modulo operator, $\mathbf{f}_j$ is the $j$-th element of vector $\mathbf{f}$ and $\boldsymbol{\Lambda} \in \mathbb{R}^{mq\times mq}$ is defined as

$$\boldsymbol{\Lambda} = \begin{bmatrix} \mathbf{A} & \mathbf{0}_{m\times m} & \cdots & \mathbf{0}_{m\times m} \\ \mathbf{0}_{m\times m} & \mathbf{A} & \ddots & \vdots \\ \vdots & \ddots & \ddots & \mathbf{0}_{m\times m} \\ \mathbf{0}_{m\times m} & \cdots & \mathbf{0}_{m\times m} & \mathbf{A} \end{bmatrix}. \tag{4}$$

Eq. (3) is a standard quadratic programming (QP) problem, which can be solved by any QP tools.

**Improve the Scalability**

Eq. (3) solves a QP problem with the computational complexity of $\mathcal{O}(m^3 q^3)$ and the storage complexity of $\mathcal{O}(m^2 q^2)$. To reduce the computational complexity and the storage complexity, we can approximately solve the original QP problem row-wisely, i.e., update the label confidence vector $\widetilde{\mathbf{f}}_j$ by fixing other variables:

$$\min_{\widetilde{\mathbf{f}}_j} \frac{1}{2}\widetilde{\mathbf{f}}_j^{\mathsf{T}} (2\mathbf{D}_{jj} + \frac{2}{\alpha})\mathbf{I}_{q\times q}\widetilde{\mathbf{f}}_j + \left( \sum_{i=1,i\neq j}^{m} \mathbf{D}_{ij}\widetilde{\mathbf{f}}_i^{\mathsf{T}} - \frac{2}{\alpha}\mathbf{p}_j^{\mathsf{T}} \right)\widetilde{\mathbf{f}}_j \tag{5}$$
$$\text{s.t. } \widetilde{\mathbf{f}}_j \mathbf{1}_q = 1, \mathbf{0}_q \leq \widetilde{\mathbf{f}}_j \leq \mathbf{y}_j.$$

By this way, the original QP problem is transformed into a series of small QP problems, and the computational complexity of the QP step is reduced from $\mathcal{O}(m^3 q^3)$ to $\mathcal{O}(mq^3)$, and the storage complexity is reduced from $\mathcal{O}(m^2 q^2)$ to $\mathcal{O}(q^2)$.

**Computational Complexity Comparison among the Regression based PLL Methods**

Table S1 compares the computational complexity between the regression based PLL methods, i.e., AGGD (TPAMI 2022), PL-CLA (JCST 2021), SDIM (IJCAI 2019), DPCLS, and DPCLS-S (Scalable DPCLS). DPCLS solves a QP problem with the computational complexity of $\mathcal{O}(m^3 q^3)$, which is the same as many SOTA PLL methods like AGGD, SDIM, and PL-CLA. DPCLS-S transforms the original QP problem into a series of smaller QP problems, and the computational complexity of the QP step is reduced from $\mathcal{O}(m^3 q^3)$ to $\mathcal{O}(mq^3)$, and the storage complexity is reduced from $\mathcal{O}(m^2 q^2)$ to $\mathcal{O}(q^2)$. This approximation solution only slightly decreases the accuracy as shown in Table S2, but largely improves the scalability.

Table S1: Computational complexity comparison between the linear regression based PLL methods.

| | AGGD | PL-CLA | SDIM | DPCLS | DPCLS-S |
|---|---|---|---|---|---|
| Computational complexity | $\mathcal{O}(m^3 + mk^3 + m^3q^3)$ | $\mathcal{O}(m^3 + m^3q^3)$ | $\mathcal{O}(m^3 + m^3q^3)$ | $\mathcal{O}(2m^3 + m^2 + m^3q^3)$ | $\mathcal{O}(2m^3 + m^2 + mq^3)$ |

Table S2: Comparison between DPCLS and DPCLS-S, where DPCLS-S indicates scalable DPCLS.

| | Glass | Ecoli | Steel | Yeast | Optdigits | Usps |
|---|---|---|---|---|---|---|
| DPCLS | **.560±.051** | **.833±.014** | **.638±.022** | **.507±.026** | **.984±.002** | **.968±.004** |
| DPCLS-S | .544±.049 | .821±.023 | .633±.022 | .496±.018 | .982±.001 | .966±.004 |

# B  Proof of Theorem 1 and Theorem 2

## B1. Proof of Theorem 1

**Definition 1.** *Denote $\mathcal{H}$ be a family of functions that map $\mathcal{X}$ to $[0,1]$ and $S = \{x_1, x_2, ..., x_m\}$ is a set of fixed samples. The empirical Rademacher complexity of $\mathcal{H}$ to set $S$ is defined as*

$$\hat{\mathcal{R}}_S(\mathcal{H}) = \frac{1}{m}\mathbb{E}\left[\sup_{h\in\mathcal{H}}\sum_{i=1}^{m}\sigma_i h(x_i)\right], \tag{6}$$

*where $(\sigma_1, \sigma_2, ..., \sigma_m)$ are Rademacher variables, and each of them is an independent uniform random variable taking value in $\{-1, +1\}$.*

The square loss function of DPCLS is $\|\mathbf{F} - \mathbf{XW}\|_F^2$. $\mathbf{F}$ can be divided into the sum of the ground-truth label matrix $\mathbf{F_G} \in \mathbb{R}^{m\times q}$ and false-positive label matrix $\mathbf{N} \in \mathbb{R}^{m\times q}$. So we can rewrite the square loss function as $\|\mathbf{F_G} + \mathbf{N} - \mathbf{XW}\|_F^2$. Based on **Definition 1**, let $\mathcal{H} = \mathbf{W} \times \mathbf{N}$ be the family of functions of DPCLS, i.e., $(\mathbf{W}, \mathbf{N}) \in \mathcal{H}$. $\ell$ is the square loss function of DPCLS, the Rademacher complexity with respective to $\mathcal{H}$ and $\ell$ can be expressed as

$$\hat{\mathcal{R}}_S(\ell \circ \mathcal{H}) = \frac{1}{m}\mathbb{E}\left[\sup_{h\in\mathcal{H}}\sum_{i=1}^{m}\sigma_i \ell(h(\mathbf{x}_i), \mathbf{F_G}_i)\right]. \tag{7}$$

Note that the square loss is $2q$-Lipschitz. According to [1], Eq. (7) is upper bounded by

$$\hat{\mathcal{R}}_S(\ell \circ \mathcal{H}) \le \frac{2\sqrt{2}q}{m}\mathbb{E}\left[\sup_{h\in\mathcal{H}}\sum_{i=1}^{m}\sum_{j=1}^{q}\sigma_{ij}(x_i\mathbf{W}_{.j} - \mathbf{N}_{ij})\right], \tag{8}$$

where $\sigma_{ij}$ is the Rademacher variable which takes value in $\{-1, 1\}$. $\mathbf{W}_{.j}$ is $j$-th column of classifier $\mathbf{W}$. Denote $\hat{\mathbf{X}} = [\hat{\mathbf{X}}_1; \hat{\mathbf{X}}_2; ...; \hat{\mathbf{X}}_q] \in \mathbb{R}^{q\times d}$, $\hat{\mathbf{X}}_q = \sum_{i=1}^{n}\sigma_{iq}x_i$. Without loss of generality, we assume the complexity of classifier $\mathbf{W}$ and the sparsity of $\mathbf{N}$ are upper bounded by $\epsilon_1$ and $\epsilon_2$ respectively, i.e., $\|\mathbf{W}\|_F \le \epsilon_1$ and $\|\mathbf{N}\|_1 \le \epsilon_2$. The right side of Eq. (8) can be relaxed as:

$$\begin{aligned}\hat{\mathcal{R}}_S(\ell \circ \mathcal{H}) &\le \frac{2\sqrt{2}q}{m}\mathbb{E}\left[\sup_{h\in\mathcal{H}}\langle\mathbf{W}^\mathsf{T}, \hat{\mathbf{X}}\rangle + \|\mathbf{N}\|_1\right] \\ &\le \frac{2\sqrt{2}q}{m}\mathbb{E}\left[\sup_{h\in\mathcal{H}}\|\mathbf{W}\|_F\left\|\hat{\mathbf{X}}\right\|_F + \|\mathbf{N}\|_1\right] \\ &\le \frac{2\sqrt{2}q}{m}\mathbb{E}\left[\sup_{h\in\mathcal{H}}\epsilon_1\left\|\hat{\mathbf{X}}\right\|_F + \epsilon_2\right].\end{aligned} \tag{9}$$

We assume each sample is normalized, i.e., $\|x_i\|_2 \le 1$, it is easy to prove that

$$\mathbb{E}_\sigma\left\|\hat{\mathbf{X}}\right\|_F^2 = \mathbb{E}_\sigma\left[\sum_{j=1}^{q}\left\|\hat{\mathbf{X}}_j\right\|_2^2\right] = \mathbb{E}_\sigma\left[\sum_{j=1}^{q}\left\|\sum_{i=1}^{m}\sigma_{ij}\mathbf{x}_i\right\|_2^2\right] \le mq. \tag{10}$$

According to Eq. (9) and Eq. (10) we have **Theorem 1**

$$\hat{\mathcal{R}}_S(\ell \circ \mathcal{H}) \le \frac{2\sqrt{2}q(\sqrt{mq}\epsilon_1 + \epsilon_2)}{m}. \tag{11}$$

## B2. Proof of Theorem 2 Inequality 1

Denote $\mathbf{F} \in \{0, 1\}^{m \times q}$ and $\mathbf{D} \in \{0, 1\}^{m \times m}$ the partial label matrix and the to be optimized semantic dissimilarity matrix. Let $\mathbf{F_G}$ and $\hat{\mathbf{D}}$ be the ground-truth label matrix and the ground-truth dissimilarity matrix. Denote $\boldsymbol{\Delta_F} = \mathbf{F_G} - \mathbf{F}$. We aim to minimize the adversarial prior between the semantic dissimilarity matrix and similarity matrix, i.e., $\left\|\mathbf{D} \odot \mathbf{FF^\top}\right\|_1$. The following inequality holds

$$\langle (\mathbf{F} + \boldsymbol{\Delta_F})(\mathbf{F} + \boldsymbol{\Delta_F})^\top, \hat{\mathbf{D}} \rangle \leq \langle \mathbf{FF^\top}, \mathbf{D} \rangle. \tag{12}$$

Expand Eq. (12), we have

$$\begin{aligned}
\langle \boldsymbol{\Delta_F}\boldsymbol{\Delta_F}^\top, \hat{\mathbf{D}} \rangle &\leq \langle \mathbf{FF^\top}, \mathbf{D} - \hat{\mathbf{D}} \rangle - \langle \mathbf{F}\boldsymbol{\Delta_F}^\top, \hat{\mathbf{D}} \rangle - \langle \boldsymbol{\Delta_F}\mathbf{F}^\top, \hat{\mathbf{D}} \rangle \\
&= \langle \mathbf{FF^\top}, \mathbf{D} - \hat{\mathbf{D}} \rangle + \langle \mathbf{F}\boldsymbol{\Delta_F}^\top, -\hat{\mathbf{D}} \rangle + \langle \boldsymbol{\Delta_F}\mathbf{F}^\top, -\hat{\mathbf{D}} \rangle \\
&\leq \|\mathbf{F}\|_F^2 \left\|\mathbf{D} - \hat{\mathbf{D}}\right\|_F + 2\|\mathbf{F}\|_F \left\|\hat{\mathbf{D}}\right\|_F \|\boldsymbol{\Delta_F}\|_F .
\end{aligned} \tag{13}$$

Due to the characteristics of PLL, at least one false-positive label exists in each PLL data set, i.e., $\|\boldsymbol{\Delta_F}\|_F \geq 1$. In this way, we have

$$\langle \boldsymbol{\Delta_F}\boldsymbol{\Delta_F}^\top, \hat{\mathbf{D}} \rangle \leq \|\mathbf{F}\|_F^2 \left\|\mathbf{D} - \hat{\mathbf{D}}\right\|_F \|\boldsymbol{\Delta_F}\|_F + 2\|\mathbf{F}\|_F \left\|\hat{\mathbf{D}}\right\|_F \|\boldsymbol{\Delta_F}\|_F . \tag{14}$$

Assume the smallest eigenvalue of $\hat{\mathbf{D}}$ is $\lambda_{\hat{\mathbf{D}}}$ and $\lambda_{\hat{\mathbf{D}}} \geq 0$, we have $\langle \boldsymbol{\Delta_F}\boldsymbol{\Delta_F}^\top, \hat{\mathbf{D}} \rangle \geq \lambda_{\hat{\mathbf{D}}} \|\boldsymbol{\Delta_F}\|_F^2$. Moreover, $\|\mathbf{F}\|_F^2$ is upper bounded by $m$ (the number of samples) and $q$ (the number of classes), i.e., $\|\mathbf{F}\|_F^2 \leq mq$. Eq. (14) can be further relaxed as

$$\lambda_{\hat{\mathbf{D}}} \|\boldsymbol{\Delta_F}\|_F^2 \leq mq \left\|\mathbf{D} - \hat{\mathbf{D}}\right\|_F \|\boldsymbol{\Delta_F}\|_F + 2\sqrt{mq} \left\|\hat{\mathbf{D}}\right\|_F \|\boldsymbol{\Delta_F}\|_F . \tag{15}$$

Let $\left\|\bar{\boldsymbol{\Delta}}_\mathbf{F}\right\|_F$ be the average distance of each sample between $\mathbf{F_G}$ and $\mathbf{F}$ (i.e., $\left\|\bar{\boldsymbol{\Delta}}_\mathbf{F}\right\|_F = \frac{1}{m} \|\mathbf{F_G} - \mathbf{F}\|_F$). Dividing $m$ on both sides of Eq. (15), we finally have

$$\left\|\bar{\boldsymbol{\Delta}}_\mathbf{F}\right\|_F \leq \frac{q}{\lambda_{\hat{\mathbf{D}}}} \left\|\mathbf{D} - \hat{\mathbf{D}}\right\|_F + \frac{2\sqrt{q}}{\lambda_{\hat{\mathbf{D}}}\sqrt{m}} \left\|\hat{\mathbf{D}}\right\|_F . \tag{16}$$

We can find that as the number of samples $m$ increases, the upper bound of $\left\|\bar{\boldsymbol{\Delta}}_\mathbf{F}\right\|_F$ decreases, which indicates that more training samples will push the partial label matrix to be close to the ground-truth one and achieve better PLL performance. Moreover, a smaller error between $\mathbf{D}$ and $\hat{\mathbf{D}}$ implies a smaller upper bound of $\left\|\bar{\boldsymbol{\Delta}}_\mathbf{F}\right\|_F$, which indicates a better dissimilarity matrix can help achieve a better label matrix.

## B3. Proof of Theorem 2 Inequality 2

Denote $\boldsymbol{\Delta_D} = \hat{\mathbf{D}} - \mathbf{D}$. According to the adversarial relationship and dissimilarity propagation of DPCLS, we assume each sample and its $k$ neighborhoods belong to the same class, then the following inequality holds

$$\langle \mathbf{F_G}\mathbf{F_G}^\top, \mathbf{D} + \boldsymbol{\Delta_D} \rangle + \mathrm{Tr}((\mathbf{D} + \boldsymbol{\Delta_D})\mathbf{L}(\mathbf{D} + \boldsymbol{\Delta_D})^\top) \leq \langle \mathbf{FF^\top}, \mathbf{D} \rangle + \mathrm{Tr}((\mathbf{DLD}^\top). \tag{17}$$

Expand Eq. (17), we have

$$\begin{aligned}
\langle \boldsymbol{\Delta_D}^\top \boldsymbol{\Delta_D}, \mathbf{L} \rangle &\leq \langle \mathbf{FF^\top} - \mathbf{F_G}\mathbf{F_G}^\top, \mathbf{D} \rangle - 2\langle \boldsymbol{\Delta_D}, \mathbf{DL} \rangle - \langle \mathbf{F_G}\mathbf{F_G}^\top, \boldsymbol{\Delta_D}^\top \rangle \\
&= \langle \mathbf{FF^\top} - \mathbf{F_G}\mathbf{F_G}^\top, \mathbf{D} \rangle + 2\langle \boldsymbol{\Delta_D}, -\mathbf{DL} \rangle + \langle \mathbf{F_G}\mathbf{F_G}^\top, -\boldsymbol{\Delta_D}^\top \rangle \\
&\leq \left\|\mathbf{FF^\top} - \mathbf{F_G}\mathbf{F_G}^\top\right\|_F \|\mathbf{D}\|_F + 2\|\mathbf{D}\|_F \|\mathbf{L}\|_F \|\boldsymbol{\Delta_D}\|_F + \|\mathbf{F_G}\|_F^2 \|\boldsymbol{\Delta_D}\| .
\end{aligned} \tag{18}$$

Assume that at least one corresponding position of $\hat{\mathbf{D}}$ and $\mathbf{D}$ has different values, i.e., $\|\boldsymbol{\Delta_D}\|_F \geq 1$, we have

$$\langle \boldsymbol{\Delta_D}^\top \boldsymbol{\Delta_D}, \mathbf{L} \rangle \leq \left\|\mathbf{FF^\top} - \mathbf{F_G}\mathbf{F_G}^\top\right\|_F \|\mathbf{D}\|_F \|\boldsymbol{\Delta_D}\|_F + 2\|\mathbf{D}\|_F \|\mathbf{L}\|_F \|\boldsymbol{\Delta_D}\|_F + \|\mathbf{F_G}\|_F^2 \|\boldsymbol{\Delta_D}\| . \tag{19}$$

Similar to **B2**, we assume the smallest eigenvalue of $\mathbf{L}$ is $\lambda_{\mathbf{L}}$ and $\lambda_{\mathbf{L}} \geq 0$, we have $\langle \mathbf{\Delta_D}^\mathsf{T} \mathbf{\Delta_D}, \mathbf{L} \rangle \geq \lambda_{\mathbf{L}} \|\mathbf{\Delta_D}\|_F^2$. $\|\mathbf{D}\|_F^2$ is upper bounded by $m$ (the number of samples), i.e., $\|\mathbf{D}\|_F^2 \leq m^2$. Since $\mathbf{F_G}$ is the ground truth label matrix, we have $\|\mathbf{F_G}\|_F^2 = m$. Eq. (19) can be further relaxed as

$$\lambda_{\mathbf{L}} \|\mathbf{\Delta_D}\|_F^2 \leq m \left\| \mathbf{F}\mathbf{F}^\mathsf{T} - \mathbf{F_G}\mathbf{F_G}^\mathsf{T} \right\|_F \|\mathbf{\Delta_D}\|_F + 2m \|\mathbf{L}\|_F \|\mathbf{\Delta_D}\|_F + m \|\mathbf{\Delta_D}\|. \qquad (20)$$

Let $\left\| \bar{\mathbf{\Delta}}_\mathbf{D} \right\|_F$ be the average distance of each corresponding position between $\hat{\mathbf{D}}$ and $\mathbf{D}$ (i.e., $\left\| \bar{\mathbf{\Delta}}_\mathbf{D} \right\|_F = \frac{1}{m^2} \left\| \hat{\mathbf{D}} - \mathbf{D} \right\|_F$). Dividing $m^2$ on both sides of Eq. (20), we finally have

$$\left\| \bar{\mathbf{\Delta}}_\mathbf{D} \right\|_F \leq \frac{1}{\lambda_{\mathbf{L}} m} \left\| \mathbf{F}\mathbf{F}^\mathsf{T} - \mathbf{F_G}\mathbf{F_G}^\mathsf{T} \right\|_F + \frac{2}{\lambda_{\mathbf{L}} m} \|\mathbf{L}\|_F + \frac{1}{\lambda_{\mathbf{L}} m}. \qquad (21)$$

Similar to **B2**, a larger number of training samples will reduce the distance between $\mathbf{D}$ and $\hat{\mathbf{D}}$, and a smaller error between $\mathbf{F}$ and $\mathbf{F_G}$ implies a smaller upper bound of $\left\| \bar{\mathbf{\Delta}}_\mathbf{D} \right\|_F$, suggesting a better dissimilarity matrix.

## C   Details of Compared Data Sets

Table S3: Characteristics of the synthetic data sets and the real-world partial label data sets, where Avg. CLs means the average size of the candidate label set.

| Type | Data set | # Examples | # Features | # Classes | Avg. CLs |
|------|----------|-----------|-----------|-----------|----------|
| Synthetic Data Set | Glass | 214 | 9 | 6 | - |
| | Steel | 1941 | 27 | 7 | - |
| | Ecoli | 336 | 7 | 8 | - |
| | Yeast | 1484 | 8 | 10 | - |
| | Optdigits | 5620 | 64 | 10 | - |
| | Usps | 9298 | 256 | 10 | - |
| | Isolet | 1559 | 617 | 26 | - |
| | Orl | 400 | 1024 | 40 | - |
| | Amazon | 1500 | 1326 | 50 | - |
| | Bookmark | 2500 | 1413 | 57 | - |
| Real-world Data Set | FG-NET | 1002 | 262 | 78 | 7.48 |
| | Lost | 1122 | 108 | 16 | 2.23 |
| | MSRCv2 | 1758 | 48 | 23 | 3.16 |
| | BirdSong | 4998 | 38 | 13 | 2.18 |
| | Malagasy | 5303 | 384 | 44 | 8.35 |
| | Soccer Player | 17472 | 279 | 171 | 2.09 |
| | Yahoo! News | 22991 | 163 | 219 | 1.91 |

We evaluated ten synthetic data sets and seven real-world partial label data sets from various domains, whose details are shown in Table S3. The real-world data sets are publicly available at `http://palm.seu.edu.cn/zhangml/` and `https://github.com/dhgarrette/low-resource-pos-tagging-2013`.

## D   Further Analysis

**Hyper-parameters Sensitivity**

Our DPCLS has four parameters, i.e., $\alpha$, $\beta$, $\lambda$ and $k$. Fig. S1 investigates their influence to DPCLS on Lost and MSRCv2. As shown in Figs. S1 (a) - (b), when $\alpha$ and $\beta$ are too large or too small, DPCLS gives relatively poor performance. DPCLS reaches best performance when $\alpha$ is selected from {0.001, 0.01} and $\beta$ is set to 0.001. Parameter $\lambda$ controls the model complexity. We can observe from Fig. S1 (c) that the proposed model performs relatively stable when $\lambda$ changes, and setting $\lambda$=0.05 is a good choice on both Lost and MSRCv2. Fig. S1 (d) indicates that the performance of DPCLS is relatively robust to different $k$.

Table S4: Comparison between DPCLS and DPCLS-T, where DPCLS-T indicates a two-stage model.

|  | FG-NET | Lost | MSRCv2 | BirdSong | Malagasy | Soccer player | Yahoo! News |
|---|---|---|---|---|---|---|---|
| DPCLS | **.077±.009** | **.770±.024** | **.557±.014** | **.751±.009** | **.676±.004** | **.532±.002** | **.626±.003** |
| DPCLS-T | .073±.018 | .712±.018● | .483±.014● | .723±.014● | .671±.005 | .530±.003 | .567±.003● |

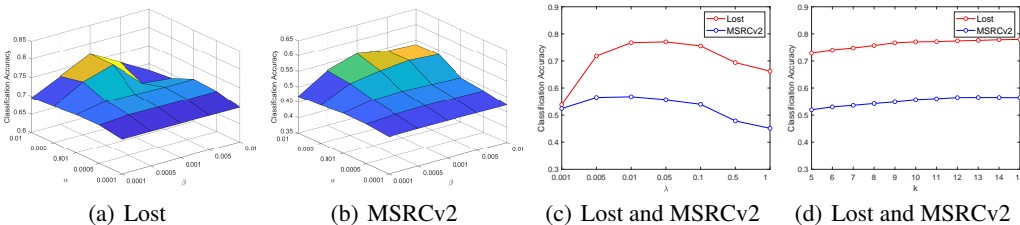

(a) Lost  (b) MSRCv2  (c) Lost and MSRCv2  (d) Lost and MSRCv2

Figure S1: Parameter sensitivity analysis for DPCLS. (a-b) Classification accuracies of DPCLS on Lost and MSRCv2 by varying $\alpha$ and $\beta$; (c) Classification accuracies of DPCLS on Lost and MSRCv2 by varying $\lambda$; (d) Classification accuracies of DPCLS on Lost and MSRCv2 by varying $k$.

## DPCLS VS. Two-stage Model

In Table S4 we compared DPCLS with a two-stage model (denoted as DPCLS-T) that performs dissimilarity matrix construction and classifier learning separately. We find that the two-stage model is significantly inferior to DPCLS, proving the advantage of end-to-end learning.

## References

[1] Andreas Maurer. A vector-contraction inequality for rademacher complexities. In Ronald Ortner, Hans Ulrich Simon, and Sandra Zilles, editors, *Algorithmic Learning Theory - 27th International Conference, ALT*, volume 9925, pages 3–17, 2016.