# OpenReview forum: "Partial Label Learning with Dissimilarity Propagation guided Candidate Label Shrinkage"
_NeurIPS.cc/2023/Conference — NeurIPS 2023 poster_

### Official Review · Reviewer_6Rie · 2023-06-22

**Soundness:** 2 fair
**Presentation:** 2 fair
**Contribution:** 2 fair
**Rating:** 6
**Confidence:** 4

**Summary:**

In this submission, authors propose a novel partial label learning method named DPCLS by realizing the effectiveness of the dissimilarity relationship. They develop a semantic similarity and dissimilarity matrix which form an adversarial relationship, which is further utilized to shrink the solution space of the label confdence matrix and promote the dissimilarity matrix.

**Strengths:**

1. Partial label learning is an interesting topic.
2. A novel partial label learning approach is proposed.
3. Experiments validate the effectiveness of the proposed approach.

**Weaknesses:**

1. The motivation is unclear and the novelty might be limited.
2. The technical details of proposed approach can be further discussed.
3. More experiments can be conducted to validate the effectiveness of the proposed approach.

**Questions:**

1. The submission does propose a partial label learning approach, but the motivation for proposing it is unclear. Why similarity and dissimilarity matrix? From this pespective, the novelty is very limited.
2. Why $l_1$ norm in Eq.(3)? Some comparative experiments?
3. The hyper-parameter $\sigma$ in Eq.(4) is not given.
4. According to Figure S1(a) and Figure S1(b) in supplementary material, the smaller the value of $\alpha$, the better the performance (with some $\beta$). Thus, it is suggested to do further ablation studies where $\lambda$, $\alpha$, $\beta$ in Eq.(6) are respectively set to zeros.
5. As shown in Algorithm 1, the proposed approach works in an iterative manner, so the convergence analysis should be conducted? How many rounds the approach will converge?
6. As shown in Table 1, the traditional baseline CLPL achieves relative superior performance while the newest baselines only achieves relative general performance? Data sets matters?

**Limitations:**

Limitations are not discussed in the current version.

---

> ### Author Rebuttal · Authors · 2023-08-09
>
> **Thank you for your time and effort in reviewing our paper.**
>
> ---
>
> **W1: The motivation is unclear and the novelty might be limited.**
>
> **Answer to W1:**
> * **Motivation**: please refer to the **Global Response** for the **Motivation** of our work, and in the final version, we will improve the introduction to better describe the motivation of our work.
> * **Novelty**: The main novelties of our work are summarized in the response to **W1** of **Reviewer ysvn**.
>
> ---
>
> **W2: Discuss the technical details of the proposed approach.**
>
> **Answer to W2:** \
> Please refer to **Q2**.
>
> ---
>
> **W3: More experiments.**
>
> Following your suggestion, we compared our method with an additional SOTA PLL method PICO [1] on the real-world data sets in **Table R1** of the **Global Response (PDF)**. PICO is designed for image classification (it involves data augmentation for images, such as image rotation, image resize, etc.), and cannot be directly evaluated on the real-world data sets (as the commonly used real-world PLL data sets are non-image data sets). To use PICO on non-image data sets, the encoder of it is changed from ResNet to a multi-layer Perceptron, and the image augmentation is changed to randomly mask 20% of the features as part of the augmented data. Compared with PICO, DPCLS achieves better performance in 5/6 cases.
>
> ---
>
> **Q1: Motivation of our paper.**
>
> **Answer to Q1:** \
> Please refer to **W1 Motivation**.
>
> ---
>
> **Q2: Why $l_1$ norm in Eq.(3)?**
>
> **Answer to Q2:**\
> We use the $\ell_1$ norm mainly because of its excellent computational property. Specifically, as both the values of the dissimilarity matrix $D$ and similarity matrix $FF^T$ are limited to [0,1], the seemingly non-smooth $\ell_1$ norm can be written as the trace of some multiplicated matrices ($\|\|D\odot F F^\mathsf{T}\|\|\_1=trace(F^TDF)$), making the associated optimization regarding $D$ and $F$ feasible. The well-known alternative like the Frobenius norm will lead to a fourth-order optimization problem, which is quite difficult to optimize. Another alternative is the $\ell_0$ norm, which is both non-convex and non-smooth. Therefore, we selected the $\ell_1$ norm . Besides, minimizing the $\ell_1$ norm fits the requirement of the adversarial prior, i.e., a larger element in $D$ indicates a smaller element in the corresponding position of $FF^T$ and vice versa.
>
> ---
>
> **Q3: The hyper-parameter $\sigma$ in Eq.(4) is not given.**
>
> **Answer to Q3:** \
> Thank you for the reminder. Following LALO [1], hyper-parameter $\sigma$ is determined by **$\sigma$=$\sum_{i=1}^{m}||x_i-x_{i k}||_2/m$**, where $x_{ik}$ denotes the $k$-th nearest neighbor of $x_i$. The setting of $\sigma$ is included in the code file. But as suggested, we will illustrate it in the last second paragraph of **Section 3**.
>
> ---
>
> **Q4: Further ablation studies.**
>
> **Answer to Q4:** \
> In **Fig. S1 (a) and (b)**, the brighter color indicates a larger value. It can be observed that when $\alpha$ ($\beta$) decreases, the classification accuracy of DPCLS drops. To better present the experimental results, we convert the image into a table format, please refer to **Table R2 and Table R3** in the **Global Response (PDF)**, which show that when $\alpha=0.01, \beta=0.001$, DPCLS achieves the best performance on data sets Lost and MSRCv2.
>
> In the following **Table RA4**, we show the performance of our model with $\alpha=0$, $\beta=0$ and $\lambda=0$, respectively (Actually, the **Table 4** of our paper has shown the case of $\alpha=\beta=0$, which is denoted as DPCLS-DP.). The results in **Table RA4** prove the effectiveness of each term of our method. (Note that the experimental results of DPCLS $\alpha=0$ and DPCLS $\beta=0$ are the same, mainly because when $\alpha=0$, the enhanced dissimilarity matrix will not affect the label confidence. And since the initial dissimilarity matrix and the similarity matrix are complementary, i.e., they are both constructed by the candidate label set, when $\beta=0$, the dissimilarity matrix will also not affect the label confidence matrix. For more details please refer to line 98 of our paper.)
>
> **Table RA4: Ablation study**
> |Data set |FG-NET| Lost |MSRCv2|BirdSong|Malagasy|
> |:----:|:----:|:----:|:----:|:----:|:----:|
> |DPCLS|.077$\pm$.009|.770$\pm$.024|.557$\pm$.014|.751$\pm$.009|.676$\pm$.004|
> |DPCLS $\lambda=0$|.047$\pm$.011|.267$\pm$.088|.110$\pm$.039|.393$\pm$.110|.245$\pm$0.11|
> |DPCLS $\alpha=0$|.073$\pm$.010|.687$\pm$.027|.466$\pm$.018|.721$\pm$.014|.612$\pm$.011|
> |DPCLS $\beta=0$|.073$\pm$.010|.687$\pm$.027|.466$\pm$.018|.721$\pm$.014|.612$\pm$.011|
>
> ---
>
> **Q5: Convergence analysis of DPCLS.**
>
> **Answer to Q5:** \
> Our algorithm solves the optimization problem in Eq. (7) of the paper with four blocks of variables, however, to the best of our knowledge, there is no general convergence proof for the IALM algorithm with more than two blocks of variables [2]. Fortunately, since each subproblem can be solved efficiently, our algorithm empirically converges well. The empirical curves can be found in the **Fig. R2** of the **Global Response (PDF)**. It is shown that our method converges within about 60 iterations.
>
> ---
>
> **Q6: Why CLPL achieves relatively superior performance.**
>
> **Answer to Q6:** \
> According to **Table 1**, CLPL achieves relatively good performance on Orl and Amazon data sets, but its performance was not as good as the newest algorithms on other data sets (See **Figure 1** and **Table 2** of the paper). Therefore, we think the observation in **Table 1** is due to the characteristics of the data sets. We will analyze this observation in **Section 5** of the final version.
>
> ---
>
> [1] 2022-ICLR-PICO: Contrastive Label Disambiguation for Partial Label Learning \
> [2] 2011-Foundations and Trends in Machine learning-Distributed optimization and statistical learning via the alternating direction method of multipliers.

---

> > ### Comment · Reviewer_6Rie · 2023-08-13
> >
> > Thanks for the response. In my opinion, this is a borderline paper. I would like to hear from other reviewers.

---

> > ### Comment · Reviewer_6Rie · 2023-08-18
> >
> > Thanks for the response. After I read all the comments of other reviewers, I would like to keep my rating for the moment.

---

> > > ### Author Response · Authors · 2023-08-18
> > >
> > > Thank you for your response and we would like to express our gratitude for your willingness to accept our paper.

---

### Official Review · Reviewer_9Zpk · 2023-07-03

**Soundness:** 3 good
**Presentation:** 3 good
**Contribution:** 3 good
**Rating:** 6
**Confidence:** 4

**Summary:**

This paper proposes a new approach to partial label learning, called DPCLS, which learns the similarity and dissimilarity matrices to improve labeling accuracy in an adversarial relationship. The proposed method is compared to several existing methods on a variety of datasets, and the results demonstrate its superior performance in most cases. The paper also includes theoretical proof of the rationality of the proposed adversarial prior and a visualization of the enhanced similarity and dissimilarity matrices on a real-world dataset.

**Strengths:**

1. This paper introduces a new partial-label learning method called DPCLS, which leverages similarity and dissimilarity matrices an adversarial relationship to effectively tackle the challenges associated with partial-label learning.

2. A theoretical proof of the rationality of the proposed adversarial prior is included to further validate the proposed method’s effectiveness.

3. Extensive experiments conducted on both real-world and artificial partial label datasets showcase the efficacy of the proposed method.


**Weaknesses:**

The paper focus on an interesting partial label learning problem and has several issues that can be improved: the datasets adopted in this paper are small-scale and the motivation of this paper should be explained in detail.

**Questions:**

1. In Fig.1 (c), compared with the co-occurring probability of 0.7, it confused me that the proposed method seems to get a better performance in the co-occurring probability of 0.8.

2. The datasets adopted in the paper are simple and small-scale which is hardly convincing, and the moderately sized datasets are suggested to adopt to validate the effectiveness of the proposed method. For example, CIFAR-100.

3. I am confused about the reported comparison results in Table 2: for the comparison method CAVL, there is a big performance drop between the reported experimental results and the CAVL paper. And can you compare the proposed method with the representative PLL method PiCO?

4. For hyper-parameters sensitivity, the author should provide the variance in the figure.

5. The motivation of the paper should be further explained in detail.


**Limitations:**

Please refer to the weakness and limitations

---

> ### Author Rebuttal · Authors · 2023-08-09
>
> **Thank you for your time and effort in reviewing our paper.**
>
> ---
>
> **W1: Small-scale data sets and motivation of this paper**
>
> **Answer to W1**
> * **Motivation**: Thank you for your valuable comments and suggestions. Please refer to **Global Response** for the detailed **Motivation** of our work, and in the final version, we will improve the introduction to better describe the motivation of our work.
> * **Data sets**: To the best of our knowledge, _the largest real-world partial label data sets_ are Soccer Player (17472 samples and 279 dimensions) and Yahoo! News (22991 samples and 163 dimensions), which were both evaluated in our paper. Moreover, some recent partial label learning works adopt some larger-scale synthetic data sets. Therefore, in this rebuttal, we further evaluated our work on CIFAR-100 (60000 samples and 1024 dimensions). Please refer to **Q2** for the detailed experimental results on CIFAR-100.
>
> ---
>
> **Q1: In Fig.1 (c), compared with the co-occurring probability of 0.7, it confused me that the proposed method seems to get a better performance in the co-occurring probability of 0.8.**
>
> **Answer to Q1:** \
> Thanks for your careful reading. Ecoli is a small-scale data set with 336 samples. Due to the influence of the random factors in producing the partial labels, the classification accuracy may fluctuate with the increase of the co-occurring probability ($\epsilon$). But, the overall trend is clear that as the co-occurring probability increases, the classification accuracy will decrease. Note that a similar phenomenon also occurs in the original papers of LALO [1], SURE [2] and AGGD [3].
>
> ---
>
> **Q2: Data sets.**
>
> **Answer to Q2:** \
> Based on your suggestion, we conducted experiments on the CIFAR-100 data set (50000 training samples and 10000 testing samples). As CIFAR-100 is not a partial label data set, following the settings of the recent works [4, 5], we randomly flipped some negative labels to constitute the candidate labels with the probability $q$ ($q$ was set to 10%, 20%, and 30%). Then, we compare our method with two state-of-the-art deep learning based PLL algorithms: PICO[4] and CAVL[5]. As CIFAR-100 is an image data set, to get a good classification on it, we need some reasonable feature representation. Therefore, we use PICO to extract the features of CIFAR-100 and reduce the dimensions of it from 1024 to 128. Then, we apply the learned representation to our model. The comparison is shown in the following **Table RA3**, where we can see that our DPCLS still achieves better performance than SOTA PLL methods on CIFAR-100. To be specific, as $q$ increases, more false positive labels exist in the candidate label sets, and PLL becomes harder. Accordingly, the accuracies of the PLL algorithms decrease rapidly, but the performance advantage of our method becomes more salient, which proves that our method is skilled in more challenging PLL tasks.
>
> ---
>
> **Table RA3: Classification accuracy (mean$\pm$std) on CIFAR-100.**
> |Data set |CIFAR-100 $q=10$%|CIFAR-100 $q=20$%|CIFAR-100 $q=30$%|
> |:----:|:----:|:----:|:----:|
> |DPCLS|**70.07**|**64.60**|**34.23**|
> |PICO|68.60|62.50|27.55|
> |CAVL|58.80|21.83|12.27|
>
> ---
>
> **Q3: Experiment results of CAVL and compared with PICO.**
>
> **Answer to Q3:**
> * **CAVL**: In the original CAVL[5] paper, **ten-fold cross-validation** was used on the real-world data sets, which consisted of 90% samples for training and 10% samples for testing. In our paper, **ten runs of 50%/50% random train/test splits** were performed on each data set, which consisted of 50% samples for training and 50% samples for testing. Different experimental settings result in different classification accuracy.
> * **PICO**: PICO [4] is designed for image classification (it involves data augmentation for images, such as image rotation, image resize, etc.), and cannot be directly evaluated on real-world data sets (the commonly used real-world PLL data sets are non-image data sets). In order to apply PICO on non-image data sets, the encoder of PICO is changed from ResNet to a multi-layer perceptron that is suitable for real-world data sets, and the image augmentation  (like image rotation, image resize, etc.) is changed to randomly mask 20% of the features as part of the augmented data. The experimental results are shown in **Table R1** of **Global Response (PDF)**. Compared with PICO, DPCLS achieves higher classification accuracy on the real-world data sets in 5/6 cases.
>
> ---
>
> **Q4: For hyper-parameters sensitivity, the author should provide the variance in the figure.**
>
> **Answer to Q4:** \
> Thank you for your suggestion, we added **standard deviation** in the **Global Response (PDF) Table R2, Table R3 and Fig. R2 (c) and (d)**.  And we will add them to the **supplementary file** in the final version.
>
> ---
> **Q5: The motivation of the paper should be further explained in detail.**
>
> **Answer to Q5:** \
> **Motivation**: Please refer to **Global Response** for detailed **Motivation**, and in the final version, we will improve the introduction to better describe the motivation of our work.
>
> ---
>
> [1] 2018-IJCAI-Leveraging Latent Label Distributions for Partial Label Learning \
> [2] 2019-AAAI-Partial Label Learning with Self-Guided Retraining \
> [3] 2022-TPAMI-Adaptive Graph Guided Disambiguation for Partial Label Learning \
> [4] 2022-ICLR-PICO: Contrastive Label Disambiguation for Partial Label Learning \
> [5] 2022-ICLR-Exploiting Class Activation Value for Partial-Label Learning

---

> > ### Comment · Reviewer_9Zpk · 2023-08-16
> >
> > Thanks for the response. I will raise my score.

---

> > > ### Author Response · Authors · 2023-08-16
> > >
> > > Thank you for your response and we would like to express our gratitude for your willingness to accept our paper.

---

### Official Review · Reviewer_vgJ9 · 2023-07-04

**Soundness:** 3 good
**Presentation:** 2 fair
**Contribution:** 3 good
**Rating:** 5
**Confidence:** 4

**Summary:**

This paper constructs a second-order similarity matrix and a semantic dissimilarity matrix. The similarity matrix is obtained by leveraging the confidence obtained from the underlying model, while the semantic dissimilarity matrix is determined based on the label candidate set and the distribution of samples in the feature space. An objective function is formulated using the adversarial relationship between the similarity matrix and the dissimilarity matrix. Additionally, the paper proposes a version of the algorithm to handle nonlinear problems by mapping the original feature space to a high-dimensional reproducing kernel Hilbert space (RKHS). Finally, the proposed method is compared with several state-of-the-art PLL algorithms and validated on ten  synthetic datasets and seven real-world datasets.

**Strengths:**

1.	An adversarial relationship is constructed for label disambiguation, and in addition, two versions are proposed: linear separable and non-linear separable.
2.	Solid theoretical analysis and ample amount of work.


**Weaknesses:**

1.	The research motivation of the paper is not clearly stated.
2.	The paper mainly focuses on describing the proposed method, without summarizing and extracting issues from existing PLL research.
3.	The principle behind the construction of the loss term based on adversarial priors is not explained.


**Questions:**

1.	What was the motivation behind the author's paper? What problems in PLL did it solve?
2.	If the relationship between two samples is characterized by both low confidence and low dissimilarity, can the loss term based on adversarial priors handle this situation?

---

> ### Author Rebuttal · Authors · 2023-08-04
>
> **Thank you for your time and effort in reviewing our paper.**
>
> ---
>
> **W1: The research motivation of the paper is not clearly stated.**
>
> **W2: The paper mainly focuses on describing the proposed method, without summarizing and extracting issues from existing PLL research.**
>
> **Answer to W1 and W2:** \
> Thank you for your valuable comments and suggestions. Please refer to the **Global Response** for the detailed review of the **Related Work** and the summarizing of their issues in the **Motivation**.  In the final version, we will improve the related works and motivation in the introduction.
>
> ---
>
> **W3: The principle behind the construction of the loss term based on adversarial priors is not explained.**
>
> **Answer to W3:** \
> The dissimilarity matrix $D$ represents the dissimilarities between samples, i.e., if the value of $D(i,j)$ is large, the sample $x_i$ is highly dissimilar to the sample $x_j$. While the similarity matrix $FF^T$ represents the similarities between samples, i.e., if the $(i,j)$-th element of $FF^T$ is large, $x_i$ is highly similar to $x_j$. Therefore, an adversarial relationship naturally exists between $D$ and $FF^T$. We, accordingly, formulate this adversarial relationship as \
>  $\mathop{\min}\limits_{D, F}\|\|D\odot F F^T\|\|\_{1}=\mathop{\min}\limits_{D, F}\sum_{i, j=1}^{m}|D_{ij} \cdot (FF^T)_{ij}|.$ \
> By minimizing the above equation, the adversarial relationship is captured and the solution space of the label confidence matrix $F$ is shrunk to achieve better label disambiguation. Besides, the quality of the dissimilarity matrix $D$ can also be enhanced. In the final version of the paper, we will explain the principle behind the adversarial prior clearer in the second paragraph of **Section 2**.
>
>
> ---
>
> **Q1: What was the motivation behind the author's paper? What problems in PLL did it solve?**
>
> **Answer to Q1:** \
> Please refer to **Global Response** for the detailed motivation of our paper and the disadvantages of the exiting PLL methods. Here we summarize the motivation of our work simply. Exploiting the information in the label space is important to label disambiguation in PLL. The current methods leverage this information by constructing a semantic dissimilarity matrix [1, 2] like SDIM [1] and PANGOLIN [2]. However, the dissimilarity matrices constructed by them are very sparse (See the **Fig 2 (e)** of the paper, the initial dissimilarity matrix is sparse). Therefore, we propose to enhance the initial dissimilarity matrix by using the geometric structure between samples in the feature space. Moreover, the dissimilarity matrix and the similarity matrix constructed by the label confidence matrix form an adversarial relationship. We use this adversarial relationship to further enhance the dissimilarity matrix and more importantly, shrink the solution space of the label confidence matrix to achieve better label disambiguation. Our model can achieve superior performance than the comparing algorithms (experiments show that our algorithm achieves significantly superior performance on real-world data sets and synthetic data sets in 79.6% cases) and obtain a better dissimilarity matrix (See the **Fig 2 (f)** of the paper, the dissimilarity matrix produced by DPCLS become denser, which is quite close to the ideal one in **Fig 2 (d)**). Besides, we also theoretically prove the effectiveness of the adversarial term in **Section 4** of the paper.
>
> ---
>
> **Q2: If the relationship between two samples is characterized by both low confidence and low dissimilarity, can the loss term based on adversarial priors handle this situation?**
>
> **Answer to Q2:** \
> Yes, our adversarial term can handle the situation when low confidence and low dissimilarity occur at the same time. Specifically, the loss of the adversarial term can be written as\
> $\mathop{\min}\limits_{D, F}\|\|D\odot F F^\mathsf{T}\|\|\_{1}=\mathop{\min}\limits_{D, F}\sum_{i, j=1}^{m}|D_{ij} \cdot (FF^T)_{ij}|$. \
> When the confidence between $x_i$ and $x_j$ is low and the dissimilarity between $x_i$ and $x_j$ is also small, the values of both $ FF^T (i,j)$ and $D(i,j)$ are small. Accordingly, the value of the above objective function is also small. Therefore, the proposed adversarial loss can accept this uncertainty case (both low confidence and low dissimilarity).  \
> On the contrary, our adversarial term does not allow two samples to have both high confidence and high dissimilarity which will lead to a large objective function value. Therefore, the proposed adversarial term adopts a conservative strategy. The reason is that PLL is a weakly supervised problem, with insufficient supervision, we should allow some uncertainty cases exist (both low label confidence and low dissimilarity). Extensive experiments have demonstrated the effectiveness of the adopted adversarial strategy. And we also theoretically prove its effectiveness in **Section 4** of this paper.
>
> ---
>
> [1] 2019-IJCAI-Partial Label Learning by Semantic Difference Maximization \
> [2] 2020-CIKE-Learning with Noisy Partial Labels by Simultaneously Leveraging Global and Local Consistencies

---

> > ### Comment · Reviewer_vgJ9 · 2023-08-15
> >
> > Thanks for the response. After reading all the reviewers' comments, most of the reviewers agreed that the paper's motivation was not clearly described, which puts the paper in a borderline state. If the AC needs to make a clear acceptance or rejection, I am inclined to give an acceptance. But for now keep the score the same (borderline accept).

---

> > > ### Author Response · Authors · 2023-08-16
> > >
> > > Thank you for your response and we would like to express our gratitude for your willingness to accept our paper.

---

### Official Review · Reviewer_ysvn · 2023-07-06

**Soundness:** 3 good
**Presentation:** 3 good
**Contribution:** 2 fair
**Rating:** 5
**Confidence:** 4

**Summary:**

The paper proposes a new method for partial label learning called Dissimilarity Propagation guided Candidate Label Shrinkage（DPCLS）. The method captures the confidence of candidate labels by constructing a constrained regression model and uses the product of the label confidence matrix and its transpose to build a second-order similarity matrix. Additionally, the method constructs a semantic dissimilarity matrix by considering the complement of the intersection of candidate label sets and propagating the initial dissimilarity relationships throughout the entire dataset using the local geometric structure of the samples. The adversarial relationship between the similarity and dissimilarity matrices is further utilized to narrow down the solution space of the label confidence matrix and facilitate the construction of the dissimilarity matrix. The method is evaluated on artificial datasets and real-world partial label datasets, demonstrating superior performance compared to existing partial label learning algorithms.

**Strengths:**

1. The paper introduces a unique combination of dissimilarity propagation and guided candidate label shrinkage for PLL, offering a fresh perspective on the problem.
2. The authors present a detailed framework, including the construction of similarity and dissimilarity matrices, leveraging local geometric structures, and extension to a kernel version, providing a comprehensive solution for PLL.
3. The proposed method is extensively evaluated on multiple artificial and real-world datasets, demonstrating its effectiveness and outperforming existing algorithms.

**Weaknesses:**

1. The structure and logic of the paper need further improvement. I recommend that the authors provide a clearer background and motivation in the introduction, enumerate the contributions of the paper, and provide an overview of the overall framework of the paper.
2. In the setting of hyperparameters of the method, the authors do not provide a specific method or principle for parameter selection. For parameters λ, α, β, and k, the authors only mention fixed values or ranges without explaining how to choose these parameters to obtain optimal performance. It is recommended that the authors provide guidance or experimental results regarding hyperparameter selection.
3. The paper has a limited number of references and lacks a comprehensive review of the latest research in the relevant field. I recommend that the authors conduct a more thorough literature search on the relevant field and provide more background and explanations of related studies in the introduction and related work sections.
4. The proposed model involves constructing similarity and dissimilarity matrices and solving the problem using the augmented Lagrange multiplier method, which may result in higher computational complexity compared to simpler approaches. It is recommended to perform complexity analysis on the algorithm.
5. In the conclusion section, the authors can further discuss the limitations of the proposed method and directions for future improvements to enhance the completeness of the conclusion.

**Questions:**

See the above.

**Limitations:**

See the above.

---

> ### Author Rebuttal · Authors · 2023-08-09
>
> **Thank you for your time and effort in reviewing our paper.**
>
> ---
>
> **W1: The structure and logic of the paper need improvement.**
>
> **Answer to W1:**
>
> Thank you for your suggestion. In the final version, we will improve the introduction to make the logic clearer. Specifically, the **Background (Related Work)** and **Motivation** of our work are presented in the **Global Response**. The enumerated contributions and the overall framework of this paper are summarized as follows.
>
> **Contribution and Novelty** of our work
> * We propagate the initial sparse semantic dissimilarity relationships to the whole data set, obtaining a dense and information-rich dissimilarity matrix.
> * We form an adversarial relationship between the enhanced dissimilarity matrix and the second-order similarity matrix constructed by the label confidence matrix, which helps shrink the solution space of the label confidence matrix.
> * The proposed model is extended to a kernel version to fit the non-linear structure of the samples and solved efficiently by an IALM-based algorithm.
> * We give some theoretical analyses and guarantees on the effectiveness of the proposed model.
> * Extensive experiments on real-world and synthetic data sets demonstrate the effectiveness of our model.
>
> **Framework** of our paper:  \
> Section 1 briefly introduces PLL, motivation and the contributions of our work. Section 2 presents the proposed model, and Section 3 shows the optimization algorithm. Section 4 illustrates the theoretical analysis of the model. Section 5 reports the experiments and the associated analyses. Finally, section 6 concludes the paper.
>
> ---
>
> **W2: Setting of hyperparameters.**
>
> **Answer to W2:** \
> **First**, determining the best hyper-parameters is a challenging task for almost all machine learning models. However, as shown in the last second paragraph of **Section 3**, most of our hyper-parameters are fixed, suggesting the robustness of our model. \
> **Second**, we determine the hyper-parameters based on the following principles. $k$ controls the number of k-nearest neighbors, and $\lambda$ controls the model complexity. As they are commonly used parameters in many related methods [1, 2], we directly followed the settings of the related works and set $k=10, \lambda=0.05$. $\alpha$ and $\beta$ introduce the adversarial term and the dissimilarity propagation term, and we set their values according to many experiments. In **Section D** of the **supplementary file**, we experimentally show their influence on the proposed model, and therefore fixed $\beta$ = 0.001 and selected $\alpha$ from {0.001, 0.01}.  \
> **Finally**, if all hyper-parameters are carefully tuned, our algorithm can be further improved. Even with these fixed hyper-parameters, our model still statistically outperforms others in 85.9% cases (85 out of 99) on the real-world data sets.
>
> ---
>
> **W3: Limited number of references and lacks a comprehensive review of the latest works**
>
> **Answer to W3:** \
> As suggested, we will enhance the references in the final version. Please refer to the **Global Response** for the comprehensive review of the related works.
>
> ---
>
> **W4: Complexity analysis.**
>
> **Answer to W4:**
> * Actually, we have already analyzed the computational complexity of our algorithm and compared it with other PLL methods in **Table S1** of the **supplementary file**. For your convenience, we paste **Table S1** here. DPCLS solves a QP problem with the computational complexity of $\mathcal{O}(m^3q^3)$, which is the same as many SOTA PLL methods like AGGD [2], SDIM [4], and PL-CLA [3].
> * We also compared the actual running time of DPCLS with other baselines in **Table RA2**, where DPCLS is only slightly slower than the baselines but with a significant accuracy improvement. More analyses can be found in the supplementary file.
>
> ---
>
> **Table S1: Computational complexity comparison between the linear regression based PLL methods.**
> |    |AGGD|PL-CLA|SDIM|DPCLS|
> |:----:|:----:|:----:|:----:|:----:|
> |Computational complexity|$\mathcal{O}(m^3+mk^{3}+m^3q^3)$|$\mathcal{O}(m^3+m^3q^3)$ |$\mathcal{O}(m^3+m^3q^3)$|$\mathcal{O}(2m^3+m^{2}+m^3q^3)$|
>
> ---
>
> **Table RA2: Accuracy Vs. Time cost**
> |Data set|Type|AGGD|PL-CLA|SDIM|DPCLS|
> |----|----|----|----|----|----|
> |Glass $r=1, \epsilon=0.8$|Accuracy|.491$\pm$.063|.458$\pm$.073|.508$\pm$.073|.560$\pm$.051|
> | |Time|2.63s|2.13s|2.25s|2.65s|
> |Ecoli $r=1, \epsilon=0.8$|Accuracy|.801$\pm$.033|.803$\pm$.031|.801$\pm$.028|.833$\pm$.014|
> | |Time|3.34s|2.88s|2.88s|3.56s|
> |Lost| Accuracy | .702$\pm$.024 |.696$\pm$.021|.736$\pm$.023|.770$\pm$.024|
> | |Time|12.22s| 5.80s| 7.26s|22.03s|
>
> ---
>
> **W5: Discuss the limitations of the proposed work**
>
> **Answer to weakness5:** \
> We agree with the reviewer that every work has some limitations. For our work, one of the major limitations is the computation issue on large-scale data sets. Although in **Section A** of the **supplementary file** (Eq. (5)), we have made the solving of the QP problem regarding the $F$-subproblem more scalable. In our work, the sizes of the dissimilarity and similarity matrices are both $m\times m$ with $m$ the number of the training samples, making the construction and the associated computation regarding them not scalable to the extreme large-scale data sets. In practice, we can remedy this issue by handling the data sets through mini-batches or an anchor graph. Nevertheless, the heavy computation burden on large-scale data sets is still a limitation of our work. In the final version of our paper, we will discuss this limitation in the **Conclusion** section.
>
> ---
>
> [1] 2018-IJCAI-Leveraging Latent Label Distributions for Partial Label Learning \
> [2] 2022-TPAMI-Adaptive Graph Guided Disambiguation for Partial Label Learning \
> [3] 2021-JCST-Partial Label Learning via Conditional-Label-Aware Disambiguation \
> [4] 2019-IJCAI-Partial Label Learning by Semantic Difference Maximization

---

> > ### Comment · Reviewer_ysvn · 2023-08-15
> > **I would like to change the rating to bordline accept**
> >
> > Thank you for your response. I would raise the scores.

---

> > > ### Author Response · Authors · 2023-08-16
> > >
> > > Thank you for your response and we would like to express our gratitude for your willingness to accept our paper.

---

### Official Review · Reviewer_sPwu · 2023-07-23

**Soundness:** 3 good
**Presentation:** 3 good
**Contribution:** 3 good
**Rating:** 7
**Confidence:** 3

**Summary:**

The manuscript delineates an innovative method, termed as DPCLS, which is designed for partial label learning. In addressing the issue  - namely label disambiguation - the DPCLS method exhibits an amalgamation of similarity relationship and dissimilarity relationship in an adversarial manner that endows the method with superior performance in comparison to baseline methods.

The manuscript manifests commendable quality through its comprehensive and well-crafted approach toward partial label learning. The clarity is evident in the lucid and well-structured exposition of the proposed methodology. The originality lies in the adversarial learning of similarity and dissimilarity relationships, addressing the challenges posed in partial label learning. The paper's significance is underscored by its superior performance over baseline methods, along with its potential to inspire future research in this domain.

**Strengths:**

S1. The DPCLS method's novelty is encapsulated in its adversarial learning manner of similarity and dissimilarity relationships, effectively addressing the unique challenges in partial label learning.

S2. The paper employs a sound theoretical approach with clear and well-structured explanations of the methodology and its components.

S3. The evaluation is comprehensive, with comparisons to baseline methods providing a compelling demonstration of the superior performance of the DPCLS method.

**Weaknesses:**

W1.  Although the paper provides a comprehensive explanation of the methodology, further technical insights regarding the implementation and specific algorithms within the DPCLS method would be beneficial. For example, does the order of Steps. 4-9 have an influence on the final performance? The auxiliary matrix $A$ needs more clarification, which is not easy to follow.

W2.  The paper falls short in providing a detailed analysis of the limitations of the proposed approach, a factor that could be significant for future research and practical applications.

W3.  Too many hyper-parameters in a method would somewhat degrade its quality.

W4.  A more detailed exposition of existing methods related to this work, including their characteristics and potential biases or weaknesses, would enrich the manuscript.

W5.  A clear motivation about real-world applications or potential application scenarios, would strengthen the practical significance.

W6.  Some claims in Section 4 Theoretical Analysis need more clarifications, e.g., the number of training samples, and dissimilarity matrix.

**Questions:**

Please find my comments on weaknesses.

**Limitations:**

Please find my comments on weaknesses.

---

> ### Author Rebuttal · Authors · 2023-08-09
>
> **Thank you for your time and effort in reviewing our paper.**
>
> ---
>
> **W1: Steps of Algorithm 1 and explanation of the auxiliary matrix $A$**
>
> **Answer to W1:**
> * **The effect of the different orders:**
> Steps 4-9 in **Algorithm 1** solve four subproblems, and the order of them will not affect the final performance empirically, as with different orders the algorithm stops only when it is converged [1, 2]. To validate this statement, we reversed the order of steps 4-7 and conducted experiments on data sets Lost, Ecoli ($e$=0.8) and Steel ($e$=0.8). The experimental results are shown in the following **Table RA1**, where we find the order of steps 4-9 will not affect the experimental results.
> * **Clarification on $A$**:
> The variable $D$ in the initial problem (Eq. (6)) has many constraints ($\mathbf{0}\_{m\times m} \leq D \leq \mathbf{1}\_{m\times m}, D_{i j}=D\_{0 i j}, \text{if} D\_{0 i j}=1$), making the $D$ subproblem very difficult to solve. To simplify the optimization, we introduce an auxiliary matrix $A$ and let $A=D$, and transfer some of the constraints on $D$ to the variable $A$. Then the initial problem in Eq. (6) equivalently becomes the problem in Eq. (7). Accordingly, the original subproblem regarding $D$ becomes two subproblems regarding $D$ (in Eq. (13)) and $A$ (in Eq. (15)), which are both easy to solve. More details can be found in the code, which has already been submitted to the appendix.
>
> As suggested, we will provide more detailed explanations about the technical insights of **Algorithm 1**.
>
> ---
>
> **Table RA1:  Origin order and reverse order classification accuracy**
> | Data set | Origin order | Reverse order|
> |:----:|:----:|:----:|
> | Lost | .770$\pm$.024 | .770$\pm$.024 |
> | Ecoli (e=0.8) |.832$\pm$.014|.832$\pm$.014|
> | Steel (e=0.8) |.638$\pm$.022|.638$\pm$.022|
>
> ---
>
> **W2:  The paper falls short in providing a detailed analysis of the limitations of the proposed approach, a factor that could be significant for future research and practical applications.**
>
> **Answer to W2:** \
> We agree with the reviewer that every work has some limitations. For our work, one of the major limitations is the computation issue on large-scale data sets. Although in **Section A** of the **supplementary file** (Eq. (5)), we have made the solving of the QP problem regarding the $F$-subproblem more scalable. In our work, the sizes of the dissimilarity and similarity matrices are both $m\times m$ with $m$ the number of the training samples, making the construction and the associated computation regarding them not scalable to the extreme large-scale data sets. In practice, we can remedy this issue by handling the data sets through mini-batches or an anchor graph. Nevertheless, the heavy computation burden on large-scale data sets is still a limitation of our work. In the final version of our paper, we will discuss this limitation in the **Conclusion** section.
>
> ---
>
> **W3. Too many hyper-parameters in a method would somewhat degrade its quality.**
>
> **Answer to W3:** \
> In the second last paragraph of **Section 3**, we have discussed how to set the hyper-parameters of our method. Specifically, our method has 4 hyper-parameters $\alpha$, $\beta$, $\gamma$ and $k$, **three of which are fixed ($\lambda, k, \beta$), and $\alpha$ is selected from {0.001, 0.01}**. As a comparison, the compared methods like SDIM [3] have two hyper-parameters, where one is selected from {0.001, 0.005, ..., 0.5} and the other one is selected from {0.00001, 0.00005, 0.0001, ..., 0.1}. Therefore, our method does not need complex parameter tuning and can achieve better classification performance. Moreover, as shown in **Fig. S1** of the **supplementary file**, the performance of our method is robust to the hyper-parameters, making it quite easy to use in practice.
>
> ---
>
> **W4: A more detailed exposition of existing methods related to this work, including their characteristics and potential biases or weaknesses, would enrich the manuscript.**
>
> **Answer to W4:** \
> Thank you for your suggestion. Please find the detailed **Related Work** in the **Global Response**. In the final version, we will add the detailed related works and discuss their characteristics.
>
> ---
>
> **W5: A clear motivation about real-world applications or potential application scenarios, would strengthen the practical significance.**
>
> **Answer to W5:** \
> The research on PLL is initially motivated by several real-world problems. For example, in video face recognition, several persons may appear on a single frame with captions indicating their names. In this case, one person is annotated by several names as the candidate labels and only one name is the correct label for this person [4]. Please refer to **Fig. R1** of the **Global Response (PDF)** for more detailed real-world applications.  \
> In fact, some data sets used in the experiments were collected from real-world scenarios, such as the data set Birdsong from the bird song classification task, FG-NET from the facial age estimation task, and Lost from the automatic face naming from videos. \
> In the final version, we will show more real-world PLL applications to strengthen the practical significance.
>
> ---
>
> **W6: Some claims in Section 4 Theoretical Analysis need more clarifications, e.g., the number of training samples, and dissimilarity matrix.**
>
> **Answer to W6:** \
> Thank you for your suggestion. $m$ denotes the number of training samples, and $D$ is the dissimilarity matrix. We will add detailed explanation for each variable in **Section 4** of the final version.
>
> ---
>
> [1] 2021-TPAMI-Partial Multi-Label Learning with Noisy Label Identification \
> [2] 2023-Tcyber-Prior Knowledge Regularized Self-Representation Model for Partial Multilabel Learning \
> [3] 2019-IJCAI-Partial Label Learning by Semantic Difference Maximization \
> [4] 2017-TKDE-Disambiguation-Free Partial Label Learning

---

> > ### Comment · Reviewer_sPwu · 2023-08-15
> >
> > Thanks for the response. I appreciate the idea of adversarial learning for similarity and dissimilarity relationships, but I also realize that the paper has some weaknesses, e.g., hyper-parameters, and its applications in the real world. Overall, I am inclined to give an acceptance.

---

> > > ### Author Response · Authors · 2023-08-16
> > >
> > > Thank you for your response and we would like to express our gratitude for your willingness to accept our paper.

---

### Author Rebuttal · Authors · 2023-08-09

Thanks to all the reviewers and the area chair for handling our paper and the valuable comments and suggestions to improve its quality. In the initial comments, we received 4 positive recommendations (1 Accept, 1 Weak Accept, 2 Borderline Accept) and 1 negative recommendation (1 Borderline Reject). In this **"Global Response"**, we will respond to two common questions posed by several reviewers, i.e., **"Related Work"** and **"Motivation"** of our work.

---

**To reviewers sPwu, ysvn and vgJ9**

**Related Work:**\
Partial label learning (PLL) [1, 2, 3] is an emerging weakly supervised learning framework. In PLL, each sample is associated with a set of candidate labels, among which only one is the ground-truth label. The key to solve the PLL problem is label disambiguation, i.e., identifying the ground-truth label of a sample from its candidate label set. We roughly divide the existing label disambiguation strategies into three categories.

The first kind of methods [4, 5, 6, 7] leverages the similarity relationship of samples in the feature space to achieve label disambiguation. For example, [4] makes predictions by weighted voting on neighboring instances. [5] used graph regularization to disambiguate the candidate labels, i.e., if two samples are similar in features, they are likely to share the same ground-truth label. [6] adopted an adaptive graph structure to estimate label confidence, and the label with the highest confidence is regarded as the ground-truth label. [7] proposed discrimination augmentation for PLL by using the class prototypes. However, when the neighboring relationships and class prototypes of samples are inaccurate, the performance of this kind of methods will be degraded.

The second kind of methods [8, 9, 10] uses the output of the model to guide label disambiguation. For example, [8] narrowed the candidate label set through a sparsity-based self-training procedure. [9] and [10] disambiguate the candidate label sets by the outputs of the deep neural network itself. However, the model output may be inaccurate (especially in the early stages of model training), which will result in performance degradation.

The third kind of methods [11, 12] uses the information of the label space to achieve disambiguation. Especially, the information of the non-candidate labels that accurately indicates a sample does not belong to that set of labels is exploited. For example, SDIM [11] first proposed to use the dissimilarity relationship of samples in the label space (if two samples do not share any common candidate labels, they must belong to the different classes), and then maximized the label confidence between dissimilar samples to achieve label disambiguation. [12] guided the disambiguation by using the dissimilarity matrix and the class prototypes simultaneously, i.e., maximize the label confidence between dissimilar samples and reduce the label confidence of those with large distances from the class prototypes. Although these methods use the information of the label space, the constructed semantic dissimilarity matrix is sparse and predefined, which limits its applicability.

---

**To reviewers ysvn, vgJ9, 9Zpk and 6Rie**

**Motivation:**\
Like SDIM [11], our method belongs to the third category. The core idea of SDIM [11] is to use the constructed semantic dissimilarity matrix $D$ to guide label disambiguation. However, the dissimilarity matrix $D$ in SDIM is sparse and predefined. Especially with a larger candidate label set, $D$ will become extremely sparse, limiting the model performance. To solve this problem, we aim to construct a denser and information-rich dissimilarity matrix to help label disambiguation. Specifically, we propose to enhance the initial dissimilarity relationships to the whole data set by the local geometric structure of samples in the feature space, i.e., If two samples $x_i$ and $x_j$ are close to each other in the feature space, their dissimilarity relationships should also be similar. When the enhanced dissimilarity matrix is obtained, another problem is how to apply it to label disambiguation. As $D\in\mathbb{R}^{m \times m}$ indicates the pairwise dissimilarity relationships among samples, we further use the label confidence matrix $F$ multiplied by its transpose to construct a second-order similarity matrix $FF^T\in\mathbb{R}^{m \times m}$ among samples. The dissimilarity matrix $D$ and the similarity matrix $FF^T$ naturally form an adversarial relationship, i.e., a larger (resp. smaller) element in $D$ implies a smaller (resp. larger) element in $FF^T$. We formulate this adversarial prior as an $\ell_1$ norm minimization problem. By optimizing it, the enhanced dissimilarity matrix $D$ can shrink the solution space of the label confidence matrix $F$ to achieve label disambiguation, and meanwhile, the similarity matrix induced from the label confidence matrix also contributes to build a better dissimilarity matrix. Theoretical analysis in **Section 4** and empirical evaluation in **Section 5** demonstrate the effectiveness of the above approach.

---

[1] 2023-NN-Partial label learning: Taxonomy, analysis and outlook \
[2] 2022-ICLR-PICO: Contrastive Label Disambiguation for PLL \
[3] 2022-ICLR-Exploiting Class Activation Value for PLL \
[4] 2012-Intelligent Data Analysis-Learning from ambiguously labeled examples\
[5] 2018-IJCAI-Leveraging Latent Label Distributions for PLL \
[6] 2022-TPAMI-Adaptive Graph Guided Disambiguation for PLL  \
[7] 2022-KDD-Partial Label Learning with Discrimination Augmentation \
[8] 2019-AAAI-Partial Label Learning with Self-Guided Retraining \
[9] 2020-ICML-Progressive Identification of True Labels for PLL  \
[10] 2021-ICML-Leveraged Weighted Loss for Partial Label Learning \
[11] 2019-IJCAI-Partial Label Learning by Semantic Difference Maximization \
[12] 2020-CIKE-Learning with Noisy Partial Labels by Simultaneously Leveraging Global and Local Consistencies

---

### Decision · Program_Chairs · 2023-09-21

**Decision:**

Accept (poster)

**Comment:**

The paper presents a dissimilarity propagation guided label shrinkage approach for partial label learning. The reviewers generally agreed that the proposed method makes a solid contribution to the addressed problem through rigorous theoretical analysis and extensive empirical study. There also exist several concerns in the reviews regarding motivation of study, usefulness of algorithm and completeness of literature survey as well. The authors managed to address some of these in their responses. After discussions, the reviewers reached a consensus that this is an interesting and technically sound paper where strengths outweigh weaknesses. We thus recommend that the paper can be accepted if space available in the program.